# OPV: Outcome-based Process Verifier for Efficient Long Chain-of-Thought Verification

## Abstract

Large language models (LLMs) have achieved significant progress in solving complex reasoning tasks by Reinforcement Learning with Verifiable Rewards (RLVR). This advancement is also inseparable from the oversight automated by reliable verifiers. However, current outcome-based verifiers (OVs) are unable to inspect the unreliable intermediate steps in the long reasoning chains of thought (CoTs). Meanwhile, current process-based verifiers (PVs) have difficulties in reliably detecting errors in the complex long CoTs, limited by the scarcity of high-quality annotations due to the prohibitive costs of human annotations. Therefore, we propose the **O**utcome-based **P**rocess **V**erifier (OPV), which verifies the rationale process of summarized outcomes from long CoTs to achieve both accurate and efficient verification and enable large-scale annotation. To empower the proposed verifier, we adopt an iterative active learning framework with expert annotations to progressively improve the verification capability of OPV with fewer annotation costs. Specifically, in each iteration, the most uncertain cases of the current best OPV are annotated and then subsequently used to train a new OPV through Rejection Fine-Tuning (RFT) and RLVR for the next round. Extensive experiments demonstrate OPV's superior performance and broad applicability. It achieves new state-of-the-art results on our held-out OPV-Bench, outperforming much larger open-source models such as Qwen3-Max-Preview with an F1 score of 83.1 compared to 76.3. Furthermore, OPV effectively detects false positives within synthetic dataset, closely align with expert assessment. When collaborating with policy models, OPV consistently yields performance gains, e.g., raising the accuracy of DeepSeek-R1-Distill-Qwen-32B from 55.2% to 73.3% on AIME2025 as the compute budget scales.

## 1 Introduction

Large language models (LLMs) have achieved remarkable performance on challenging reasoning tasks (OpenAI et al., 2024; DeepSeek-AI et al., 2025; Yang et al., 2025; OpenAI et al., 2025). This advancement is largely attributed to the growing use of verifiable oversight. Verifiers are crucial components that not only assign rewards in Reinforcement Learning with Verifiable Rewards (RLVR) (Lambert et al., 2024) but also select optimal responses in test-time scaling (Zhang et al., 2025a) and benchmark the capabilities of LLMs (Hendrycks et al., 2021; He et al., 2024). As LLMs generate increasingly long and intricate chains of thought (CoTs), the fidelity of these verifiers becomes a critical factor that determines the capability and reliability of LLMs.

Existing verifiers fall into two categories, each with its own limitations. Outcome-based verifiers (OVs) assess only the final answer against the ground truth, and overlook the reliability of intermediate steps in long CoTs. In contrast, process-based verifiers (PVs) (Lightman et al., 2023) examine the entire CoT step-by-step to locate errors. However, they struggle with complex reasoning structures in the input CoT and incur prohibitive costs in both automated verification and expert annotations. Previous works (Wang et al., 2023; Luo et al., 2024; Zhang et al., 2025b) resort to coarse heuristics for training and fail to provide accurate correctness verdicts or error locations. This landscape highlights the need for a more accurate and efficient paradigm for long CoT verification.

---

† Corresponding author

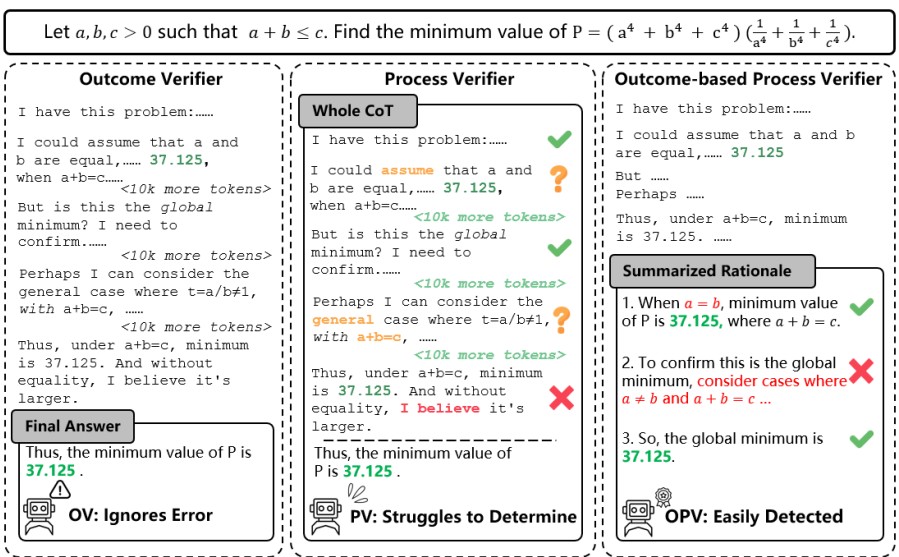

Figure 1: Comparison of different verification paradigms. The policy model happens to guess the correct answer, but fails to solidly prove it, resulting in flawed reasoning. **Left:** The Outcome-based Verifier ignores underlying reasoning failures. **Middle:** The Process-based Verifier examines the complex thinking process step-by-step, struggling to identify the tricky logic dependencies. **Right:** The **O**utcome-based **P**rocess **V**erifier efficiently detects potential process errors from the summarized rationale.

To this end, we propose the **O**utcome-based **P**rocess **V**erifier (OPV), a process verifier that operates on summarized outcomes from long CoTs (Fig. 1). Analogously to summarization, our approach first preserves only the key steps that contribute to the final answer and discards redundant components (e.g., trial-and-error attempts, recalculations, self-overturned assumptions) to form a concise solution path. Then it performs step-by-step verification on the summarized outcome and presents a correctness verdict and, if incorrect, the error location. Compared with OVs, OPV provides more fine-grained supervision, which is useful for the policy model. Moreover, its summarization process significantly reduces the complex, redundant reasoning structures in the input CoT, making OPV more efficient and less susceptible to interference from redundancy than vanilla PV. The simplified CoT also facilitates human annotation, which allows for collecting large-scale, fine-grained expert annotations for training.

To streamline massive expert annotation, we adopt an iterative human-in-the-loop framework driven by active learning (Fig. 2). In each round, the current OPV evaluates each summarized solution multiple times and selects the most uncertain cases for annotation. Expert annotators then provide natural language explanations, correctness verdicts, and error localizations. The newly annotated data are incorporated to retrain the OPV using a combination of off-line rejection fine-tuning and on-line reinforcement learning. This strategy effectively strengthens the verifier by focusing on its weaknesses under limited annotation budgets. After several iterations, we curated 40k annotated solutions across diverse domains of problems spanning from K-12 to undergraduate levels, including a high-quality held-out evaluation set of 2.2k sample answers, namely the OPV-BENCH.

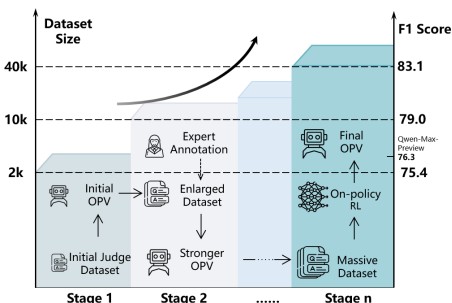

Figure 2: Our iterative framework yields enlarged judgment dataset and improved verifier performance.

Empowered by active learning with fine-grained supervision, OPV demonstrates strong performance and broad applicability. Despite its compact size, it achieves performance comparable to much larger open-source models across multiple public and internal benchmarks, including our held-out

OPV-BENCH. We further validate OPV's versatility in assisting reasoning models across multiple stages, from training to inference. On AM-DeepSeek-R1-0528-Distilled (Ji et al., 2025), a widely used synthetic dataset verified solely by final answers, OPV identifies false positives at an estimated rate of 7.0%, closely aligning with expert assessments. In collaboration with various policy models, OPV consistently enhances their test-time performance, with the improvement margin growing as the compute budget scales. For instance, it boosts the accuracy of DeepSeek-R1-Distill-Qwen-32B from 55.2% to 73.3% on AIME2025.

## 2 METHOD

We define the Outcome-based Process Verifier (OPV), a novel framework that bridges outcome and process verification through a faithful, verifiable proxy for long CoT verification (§ 2.1). To build a competent OPV model, our approach leverages an iterative active learning framework to select high-quality data for expert annotations (§ 2.2) and a combination of off-line and on-line learning approaches to update the verifier(§ 2.3). We also illustrate the statistics of the annotated data (§ 2.4).

### 2.1 TASK FORMULATION

Recent reasoning LLMs generate long chains of thought (CoTs) to solve mathematical problems. These CoTs consist of numerous sequential steps with complex inter-step dependencies, making verification particularly challenging. Two dominant verification paradigms have emerged.

• **Outcome-based Verifier (OV)** checks only the final answer against the ground truth.

• **Process-based Verifier (PV)** sequentially verifies each step throughout the whole CoT.

Both approaches have limitations. OV suffers from false positives — accepting correct answers derived from flawed reasoning — and cannot pinpoint errors in incorrect solutions. PV, with its fine-grained nature, struggles with intricate dependencies of long CoTs. Moreover, verifying lengthy CoTs is computationally expensive for verifier models and labor-intensive for possible human annotators.

To bridge this gap, we propose the **Outcome-based Process Verifier (OPV)**, a hybrid paradigm that balances faithfulness and efficiency. OPV first summarizes verbose and meandering CoT trajectories into concise, linear solution paths, retaining only the key steps that contribute to the final result while pruning redundant explorations. This distilled summary serves as a faithful proxy of the underlying reasoning rationale, enabling both efficient verification and large-scale human annotation. The verifier then performs a step-by-step validation on this summary to identify the first erroneous step.

Given a CoT generated for a problem $P$, a summarizer is first applied to produce a structured, $n$-step solution $\mathcal{S} = \{s_0, \ldots, s_{n-1}\}$. Subsequently, our OPV, denoted as $\pi$, takes the problem and the structured solution as input and predicts the index $\hat{\ell}$ of the first incorrect step, together with a natural language explanation $\hat{\mathcal{E}}$:

$$(\hat{\mathcal{E}}, \hat{\ell}) \sim \pi(\cdot \mid P, \mathcal{S}), \quad \hat{\ell} \in \{-1, 0, \ldots, n-1\} \tag{1}$$

Here, $\hat{\ell} = -1$ indicates a fully correct solution. We focus on identifying the first error, as subsequent steps — though potentially valid in isolation — are built upon faulty premises and thus lack mathematical soundness.

### 2.2 ACTIVE LEARNING FRAMEWORK

Finding potential errors within an answer, even after summarization, is a challenging task. Therefore, it is essential to maximize the utilization of human annotation. To achieve this, we have constructed an iterative human-in-the-loop active learning framework, as shown in Fig. 3. We start with our base verifier $\pi_0$. In each round, we first use our best OPV model to identify the most uncertain cases for annotation. After human annotation, we then use a combination of offline expert iteration and online reinforcement learning to maximally utilize the information obtained from annotation.

**Data Preparation.** Initially, we constructed a large data pool of high-quality problems and to-be-verified solutions sampled from top-tier models, represented by the unlabeled data pool $\mathcal{D}_{\mathcal{U}} = \{(P_i, \mathcal{S}_i)\}$. All sampled solutions are pre-summarized by DeepSeek-V3 (DeepSeek-AI et al., 2024)

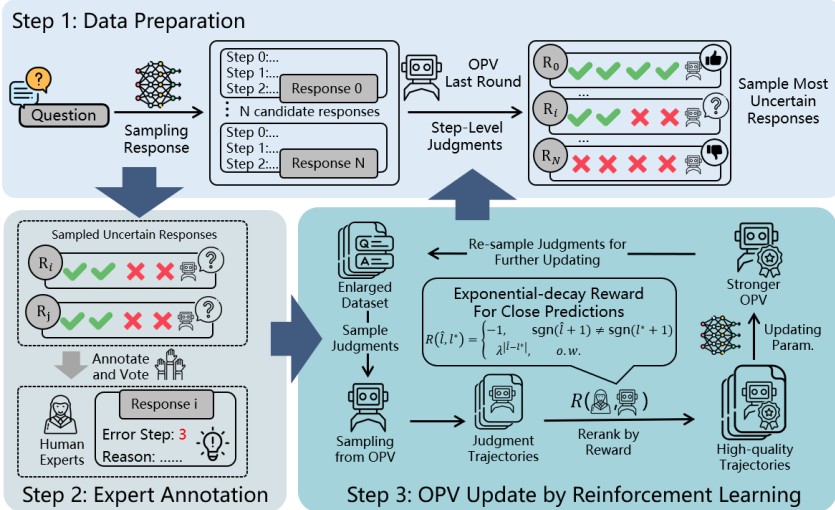

Figure 3: **Overview of our iterative active learning framework. Step 1:** For each sampled solution from the data pool, the verifier generates $N$ candidate judgment trajectories and constructs a training set selected by uncertainty scores. **Step 2:** Human experts annotate the most uncertain solutions with ground truth error positions and explanations, expanding the dataset for OPV refinement. **Step 3:** We then sample high-quality trajectories from the annotated data. The current OPV is then updated using these trajectories with off-line or on-line algorithms.

to preserve core rationale. Our set of annotation labels $\mathcal{D}_{\mathcal{L}_0}$ is empty at annotation round $0$. We also established the annotation protocol for our annotators. The complete details are available in Appendix A.1

**Selection Strategy.** Since verifiers exhibit uneven competency across different types of mathematical reasoning errors, random sampling wastes resources on cases already handled well. We therefore select the most uncertain cases at each round $t$. For each unlabeled pair $(P_i, \mathcal{S}_i) \in \mathcal{D}_{\mathcal{U}}$, the current OPV $\pi_t$ performs $N$ independent verifications to obtain a set of predicted indices $(\hat{\mathcal{E}}_i^{(j)}, \hat{\ell}_i^{(j)})$. where $\hat{\mathcal{E}}^{(j)}$ and $\hat{\ell}_i^{(j)}$ are the explanation and index predicted by the $j$-th roll-out.

To quantify the model's uncertainty, we compute a consistency score based on the frequency of the most common prediction:

$$\text{Consistency}(\mathcal{S}_i) = \frac{1}{N} \max_{\ell} \sum_{j=1}^{N} \mathbb{I}\left[\hat{\ell}_i^{(j)} = \ell\right] \quad (2)$$

A lower consistency score indicates higher uncertainty due to disagreement among the verifier's predictions. We identify the most uncertain cases:

$$\mathcal{Q}_t = \{(P_i, \mathcal{S}_i) \in \mathcal{D}_{\mathcal{U}} \mid \text{Consistency}(\mathcal{S}_i) < \tau_t\} \quad (3)$$

where $\tau_t \in (0, 1)$ is a dynamically adjustable threshold controlling annotation cost. To mitigate possible overconfidence, we also sample a small proportion of high-consistency data.

**Human Annotation.** We then send the selected data $\mathcal{Q}_t$ for expert annotation. Expert annotation yields new labeled data entries with ground truth error positions $\ell_i^*$ and reasons $\mathcal{E}_i^*$ to enlarge the annotation set:

$$\mathcal{D}_{\mathcal{L}t+1} \leftarrow \mathcal{D}_{\mathcal{L}t} \cup \{(P_i, \mathcal{S}_i, \mathcal{E}_i^*, \ell_i^*)\}_{(P_i, \mathcal{S}_i) \in \mathcal{Q}_t} \quad (4)$$

We then train on the new dataset $\mathcal{D}_{\mathcal{L}t+1}$ to obtain a stronger OPV model $\pi_{t+1}$.

## 2.3 OPV Update by Reinforcement Learning

Human annotation only provides the index of the error with a concise explanation, not a thorough reasoning trajectory that checks each step in detail. Therefore, we use a combination of online and offline approaches to refine the OPV model after each round of annotation. The updated model is then used again to select data for the next round of annotation.

**Expert Iteration.** For each annotated data entry $(P, \mathcal{S}, \mathcal{E}^*, \ell^*)$, we sample multiple verification attempts from the current OPV. We also sample from other powerful models such as those in the R1 and Qwen families to further boost performance. We retain only those generated verifications that are most consistent with the annotation (i.e., where the predicted index $\hat{\ell}$ matches the ground truth $\ell^*$). These valid verification trajectories are then added to the global verification dataset to update the OPV model. Following expert iteration (Anthony et al., 2017), we iterate this process to maximize performance gains.

**On-line Reinforcement Learning.** We also use online reinforcement learning approaches to stimulate the verification ability. To ensure stable online RL training, we filter the dataset by excluding: (1) highly ambiguous cases that challenge even experts, avoiding annotation noise; and (2) trivial cases with obvious errors, preventing bias toward oversimplified patterns. Then, given a model-predicted index $\hat{\ell}$ and ground-truth index $\ell^*$, we define the exponential-decay reward:

$$R(\hat{\ell}, \ell^*) = \begin{cases} -1 & \text{if } \mathrm{sgn}(\hat{\ell} + 1) \neq \mathrm{sgn}(\ell^* + 1), \\ \lambda^{|\hat{\ell} - \ell^*|} & \text{otherwise} \end{cases}$$

where $\lambda \in (0, 1)$ controls the penalty for localization errors. The reward is strongly negative only when misclassifying correctness (correct as incorrect or vice versa). Otherwise, it remains positive with exponential decay based on distance error. This design addresses the sparse reward problem for challenging samples requiring precise error localization. We then adopt the DAPO algorithm Yu et al. (2025) to obtain our final OPV model.

## 2.4 Final Dataset Statistics

Our framework improves the OPV model's performance while simultaneously establishing a massive, high-quality dataset. After completing all rounds of the "annotate-then-train" process, we progressively scale up the dataset to over 40k expert annotations and over 80k high-quality judge trajectories. The final fully annotated set spans multiple difficulty levels and knowledge domains. Particularly, we meticulously curate high-quality samples for evaluation and construct a held-out evaluation set of 2.2k samples, namely the OPV-Bench, to effectively estimate the verification ability of the trained OPV model. The detailed breakdown of all annotated data and the OPV-Bench is available at Appendix A.2.

# 3 Experiment

## 3.1 Experiment Setup

**Implementation.** In our framework, we use the R1-Distill-Qwen-32B (DeepSeek-AI et al., 2025) model to fine-tune the OPV. Further implementation details can be found in Appendix B.

**Evaluation.** We evaluate model performance on OPV-Bench and ProcessBench (Zheng et al., 2024) using three distinct correctness criteria with varying levels of stringency. The *precise* criterion requires exact identification of the erroneous step for a judgment to be considered correct. The *approximate* criterion adopts a more error-tolerant approach, accepting predictions as correct when the identified step is adjacent to the actual error position. Finally, the *rough* criterion considers any error detection for an incorrect answer as a correct judgment. Under each criterion, we compute accuracy separately for erroneous and correct samples, and calculate the harmonic mean of precision and recall of correct samples to obtain the F1 score.

We then compare the performance of OPV with various state-of-the-art open-source models, including the widely-used Deepseek-R1-0528, Qwen3-Max-Preview, and gpt-oss-120b. We apply the same prompt engineering approach to repurpose these models as critic models. The prompt template is provided in Appendix C. To underscore the superior quality of our expert annotations compared to heuristic labeling, we also evaluate Qwen2.5-Math-PRM-72B, a discriminative process reward model trained on labels that integrate Monte Carlo estimation with LLM-as-a-judge. To better demonstrate the effectiveness of our training framework, we also include several intermediate models (OPV-Stage1 and OPV-Stage2) for comparison. These models have undergone different rounds of annotate-then-train iterations and are trained on different amounts of annotated data, as shown in Fig. 2. The final OPV model is obtained after the final stage.

## 3.2 BENCHMARK EVALUATION

Table 1: Evaluation results on ProcessBench and OPV-Bench. We report accuracy and F1 scores for three evaluation standards: precisely/absolutely identifying erroneous steps, approximately identifying erroneous steps (within ±1 steps), and roughly identifying whether the whole solution contains errors. All results are reported under *maj@8* voting.

| Model | Precise/Abs. Accuracy | Precise/Abs. F1 | Approximate Accuracy | Approximate F1 | Rough Accuracy | Rough F1 |
|---|---|---|---|---|---|---|
| *ProcessBench (With Standard Answers)* | | | | | | |
| Qwen3-Max-Preview | 83.2 | 78.9 | 89.4 | 85.5 | 95.4 | 93.2 |
| DeepSeek-V3-0324 | 74.5 | 71.7 | 83.7 | 79.9 | 94.4 | 92.0 |
| DeepSeek-R1-0528 | 82.2 | 77.7 | 88.7 | 84.6 | 95.4 | 93.1 |
| gpt-oss-120b (high) | 83.3 | 78.8 | 89.2 | 85.1 | 95.7 | 93.5 |
| Qwen2.5-Math-PRM-72B | 77.2 | 74.0 | 84.9 | 81.2 | 93.7 | 91.1 |
| R1-Distill-Qwen-32B | 64.7 | 60.8 | 73.6 | 67.5 | 87.4 | 81.3 |
| OPV-32B | 80.1 | 76.2 | 87.6 | 83.8 | 96.2 | 94.4 |
| *ProcessBench (Without Standard Answers)* | | | | | | |
| Qwen3-Max-Preview | 84.4 | 79.9 | 89.8 | 85.9 | 95.2 | 92.9 |
| DeepSeek-V3-0324 | 72.1 | 69.8 | 81.0 | 77.2 | 91.5 | 88.4 |
| DeepSeek-R1-0528 | 83.2 | 79.3 | 89.3 | 85.8 | 96.0 | 94.1 |
| gpt-oss-120b (high) | 84.7 | 80.3 | 90.4 | 86.6 | 96.2 | 94.2 |
| Qwen2.5-Math-PRM-72B | 76.7 | 73.2 | 83.9 | 79.8 | 91.2 | 87.8 |
| R1-Distill-Qwen-32B | 75.8 | 73.6 | 83.6 | 80.5 | 93.3 | 91.1 |
| OPV-32B | 80.9 | 76.8 | 88.1 | 84.1 | 95.8 | 93.8 |
| *OPV-Bench(With Standard Answers)* | | | | | | |
| Qwen3-32B | 68.0 | 73.4 | 72.4 | 76.2 | 77.8 | 79.9 |
| QwQ-32B | 65.2 | 71.6 | 70.3 | 74.8 | 75.6 | 78.2 |
| DeepSeek-V3-0324 | 67.9 | 71.7 | 72.9 | 75.0 | 78.6 | 79.1 |
| DeepSeek-R1-0528 | 67.0 | 71.2 | 72.3 | 74.6 | 78.3 | 79.0 |
| Qwen3-Max-Preview | 66.4 | 71.2 | 71.7 | 74.6 | 78.0 | 79.0 |
| gpt-oss-120b (high) | 61.0 | 66.6 | 69.8 | 72.1 | 77.7 | 77.8 |
| Qwen2.5-Math-PRM-72B | 55.4 | 66.5 | 58.4 | 70.0 | 66.0 | 72.3 |
| R1-Distill-Qwen-32B | 70.5 | 76.3 | 74.9 | 79.1 | 78.7 | 81.7 |
| OPV-Stage1 | 68.8 | 75.7 | 72.6 | 78.0 | 76.1 | 80.2 |
| OPV-Stage2 | 71.6 | 75.9 | 76.2 | 79.0 | 81.1 | 82.6 |
| OPV-32B | 78.9 | 79.1 | 83.3 | 82.7 | 87.2 | 86.2 |
| *OPV-Bench(Without Standard Answers)* | | | | | | |
| Qwen3-32B | 61.2 | 68.7 | 67.0 | 72.1 | 74.2 | 76.7 |
| QwQ-32B | 58.2 | 66.7 | 64.2 | 70.1 | 70.9 | 74.2 |
| DeepSeek-V3-0324 | 60.8 | 67.8 | 66.0 | 70.8 | 72.8 | 75.2 |
| DeepSeek-R1-0528 | 56.9 | 64.7 | 63.2 | 68.2 | 71.7 | 73.6 |
| Qwen3-Max-Preview | 61.0 | 67.3 | 67.0 | 70.8 | 75.2 | 76.3 |
| gpt-oss-120b (high) | 57.9 | 64.1 | 66.8 | 69.3 | 75.6 | 75.4 |
| Qwen2.5-Math-PRM-72B | 55.1 | 66.0 | 58.2 | 67.6 | 63.4 | 70.4 |
| R1-Distill-Qwen-32B | 61.7 | 71.1 | 65.0 | 72.9 | 69.4 | 75.5 |
| OPV-Stage1 | 60.9 | 70.3 | 64.8 | 72.4 | 69.8 | 75.4 |
| OPV-Stage2 | 64.4 | 70.3 | 70.1 | 73.8 | 77.6 | 79.0 |
| OPV-32B | 71.9 | 74.7 | 78.2 | 79.1 | 83.1 | 83.1 |

Our experiments on two benchmarks reveal distinct challenges in evaluating reasoning verifiers. Tab. 1 presents results on both PROCESSBENCH and OPV-BENCH, highlighting their different characteristics and difficulty levels.

PROCESSBENCH, which samples answers from LLMs without thinking capabilities, exhibits performance saturation. As shown in the "Rough F1" column, most long reasoning models successfully detect the *existence* of errors in the reasoning process, achieving F1 scores above 90%. This saturation indicates that error patterns in PROCESSBENCH are more readily identifiable and less representative of sophisticated reasoning failures. The limitation arises because models without thinking mechanisms typically produce more elementary errors, whereas newer thinking LLMs generate more nuanced and subtle logical flaws. Moreover, PROCESSBENCH exclusively comprises problems with explicit, verifiable outcomes, which are more straightforward compared to proof-based problems that demand complex multi-step reasoning.

In contrast, OPV-BENCH presents significantly more complex challenges to verifiers. The test set encompasses a wider spectrum of problems and requires more advanced skills to identify exact errors. Our iterative training paradigm demonstrates its effectiveness by boosting a 32B model's performance above much larger models. Notably, most open-source reasoning models struggle with identifying error positions (see Appendix D for detailed breakdown). While these models achieve high recall, their poor precision indicates an inability to effectively detect errors in solutions—a limitation possibly inherited from training solely on verifiable outcomes.

## 4 APPLICATION

We further explore various applications of OPV in this section, demonstrating how it facilitates both the training and inference phases of LLM development.

### 4.1 EXAMINING AM-DEEPSEEK-R1-0528-DISTILLED USING OPV

A primary application of OPV is providing fine-grained supervision for robust training. Outcome-verified synthetic datasets often contain false positives. By using OPV for process verification, we can identify and remove these instances, yielding higher-quality datasets for supervised fine-tuning.

To validate this approach, we evaluated AM-DeepSeek-R1-0528-Distilled using OPV. Each data entry was verified 8 times, with entries flagged as problematic if OPV reported errors $\geq 6$ times. We conducted human evaluation on 50 randomly sampled problems to check the reliability of OPV under this setting. Out of 674k math-related data entries checked, 53.7k were flagged as problematic by OPV. The distribution of OPV votes and human evaluation results are available at Fig. 4 and Tab. 2. Human evaluation results show that OPV demonstrates high reliability in verification, with 88% of the judgments being valid. Therefore, it is estimated that more than $\frac{53.7}{674.0} \times 88\% = 7.0\%$ data entries checked contain process errors.

For cost efficiency, we used vanilla summaries rather than re-summarizing the thinking content. However, some answers might be incorrectly flagged due to inappropriate summarization. We note that 2 of the 50 solutions checked (marked as "Poor Summary") were actually correct but identified as incorrect by OPV because their original summaries introduced logical gaps. This again highlights the importance of re-summarization for precise verification.

### 4.2 SCALING OF COLLABORATIVE REASONING

Beyond evaluating our OPV on static benchmarks, we study whether it can improve test-time performance in collaboration with policy models. In a collaborative setting, a policy model first samples $N$ complete solutions for a given problem. The verifier then checks each solution $M$ times to estimate its correctness. Finally, we select the answer by aggregating verification verdicts across all solutions.

We conduct experiments on AIME2025 and consider both moderate-sized distilled models and top-tier models as policies. We use OPV as the verifier and define the verification pass rate as the proportion of runs in which the verifier deems the solution correct. We set $N = 8$ and $M = 16$. When multiple answers tie with the same frequency, we report the average accuracy across the tied

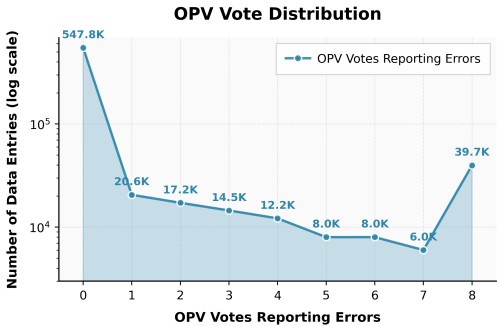

Figure 4: The distribution of OPV's votes on AM-DeepSeek-R1-0528-Distilled. 0 means OPV does not report any error among the 8 votes.

| Error Category | Count | % |
|---|---|---|
| Logical Process Errors | 24 | 48.0 |
| Non-Math Problem | 1 | 2.0 |
| Hallucination | 19 | 38.0 |
|     Problem Modification | 10 | 20.0 |
|     Added False Conditions | 9 | 18.0 |
| Correct | 6 | 12.0 |
|     Poor Summary | 2 | 4.0 |
|     Fully Correct | 4 | 8.0 |
| **Total** | **50** | **100.0** |

Table 2: Error Analysis by Human Experts

Table 3: Performance of different policies and collaborative reasoning strategies on AIME2025. **Pass@1**: single-sample accuracy averaged over all samples. **Pass@8**: oracle success with up to eight samples (correct if any of the eight is correct), serving as an upper bound for 8-sample strategies.

| Policy | Pass@1 | Majority Voting@8 | Best-of-8 | Verifier Voting@8 | Pass@8 |
|---|---|---|---|---|---|
| DeepSeek-R1-Distill-Qwen-7B | 39.1 | 49.2 | **56.3** | 55.4 | 66.7 |
| DeepSeek-R1-Distill-Qwen-32B | 55.9 | 63.8 | 66.6 | **68.0** | 76.7 |
| DeepSeek-R1-Distill-Llama-70B | 44.0 | 50.0 | 56.1 | **57.8** | 70.0 |
| DeepSeek-R1-0528 | 87.1 | 88.3 | 87.5 | **90.8** | 96.7 |
| gpt-oss-120b | 92.3 | 93.3 | 95.1 | **96.7** | 96.7 |
| QwQ-32B | 70.0 | 78.3 | 76.2 | **80.0** | 83.3 |
| Qwen3-235B-A22B-Thinking-2507 | 95.4 | 96.7 | 97.9 | **98.3** | 100.0 |

answers. Notably, we directly employ the summary part of the original CoT here to reduce compute budget and avoid introducing extra noise. We evaluate the following collaborative strategies and compare them against verifier-free majority voting:

**Majority Voting.** Among the $N$ sampled solutions, choose the most frequent answer.

**Best-of-$N$.** Rank solutions by their verification pass rate and output the answer from the top-ranked solution.

**Verifier Voting.** Use the verification pass rate as the weight of each solution and select the answer with the highest verifier-weighted frequency.

As shown in Tab. 3, OPV consistently boosts performance across all policies in the collaborative setting. For distilled models, verifier voting yields substantial gains of 6.1% on average over majority voting. Even for top-tier models where majority voting already achieves high accuracy, voting with OPV still provides stable improvements of 2.3% on average. Notably, verifier voting matches Pass@8 (the 8-sample oracle) for gpt-oss-120b. Moreover, verifier voting outperforms Best-of-8 on most policies, indicating that aggregating verification pass rates across multiple candidates is generally more robust than selecting a single best solution.

We further evaluate how collaborative reasoning scales with the policy sampling size $N$ and the verifier sampling size $M$. The evaluation focuses on DeepSeek-R1-Distill-Qwen-32B as

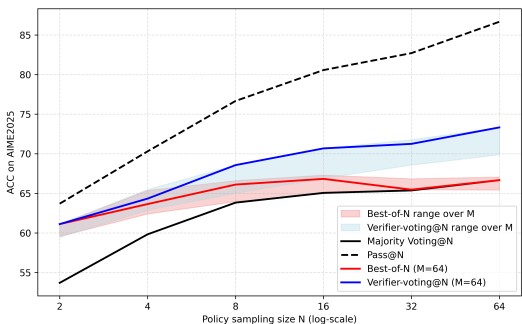

Figure 5: Performance of collaborative reasoning scaling along policy sampling size $N$ and verifier sampling size $M$. Colored bands denote the area covered by accuracy curves as $M$ increases.

the policy, scaling N and M from 1 to 64, respectively. Initial sampling uses $N = 64$ and $M = 64$. For each $(N, M)$ configuration, random subsets are chosen to evaluate different strategies' performance, with 64 repetitions to determine average scores.

Fig. 5 demonstrates that accuracy improves with larger N across all strategies, with OPV-enabled Best-of-N and Verifier-Voting consistently outperforming Majority-Voting. Verifier-Voting achieves the highest performance, reaching $73.3\%$ at $N = 64$ and $M = 64$ — a 6.7 point improvement over Majority-Voting. Best-of-N gradually converges to Majority-Voting as N increases, indicating that integrating OPV's verification with voting mechanisms yields superior overall performance. These results confirm that OPV collaborates effectively with policy models, with gains increasing proportionally to larger computational budgets. Besides, although the policy model's native summary part is more concise than the summarized rationale used during training, OPV still proves effective, which can be attributed to the annotation protocol's tolerance for minor logical gaps.

## 5 RELATED WORK

**LLM Reasoning.** Reasoning is regarded as a core capability toward artificial general intelligence. LLMs predominantly perform reasoning in a chain-of-thought (CoT) manner (Wei et al., 2022). Frontier LLMs (OpenAI et al., 2024; DeepSeek-AI et al., 2025) further extend the length and complexity of their CoTs to solve challenging math problems. Rejection Fine-Tuning (RFT) (Yuan et al., 2023) and Reinforcement Learning with Verifiable Rewards (RLVR) (Lambert et al., 2024) rely on verifiers to curate high-quality training data. Recent works like OREAL (Lyu et al., 2025) underscore the importance of well-designed verifiers in Reinforcement Learning (RL). Motivated by these observations, we develop a verifier that generates verification trajectories in a CoT manner and a framework that iteratively updates it using RFT and RL.

**Outcome-based verification v.s. Process-based verification.** Outcome-based verifiers (OVs) assess solutions solely by whether the final answer matches the ground truth. Rule-based verifier like the Math-Verify library from HuggingFace[1] and LLM-as-Judge like CompassVerifier (Liu et al., 2025) make simple answer check scalable. However, OVs overlook the reliability of intermediate steps. In contrast, Processed-based Verifier (PV) meticulously verify the reasoning process step-by-step. Empirical evidence shows that PVs can outperform OVs on difficult mathematical benchmarks (Lightman et al., 2023). Yet the prohibitive cost of fine-grained process annotation poses a practical bottleneck. Prior work has therefore resorted to coarse heuristic for training. Wang et al. (2023); Luo et al. (2024) use Monte Carlo Method to assess and assign credit to intermediate steps, which can introduce simulation bias and label noise. Zhang et al. (2025b); Yang et al. (2025); She et al. (2025); Duan et al. (2025) use stronger teacher models as judge to provide step-level labels, though the resulting verifiers are ultimately bounded by the teachers' capacity. With the rapid progress of LLM reasoning, Zhang et al. (2024); Mahan et al. (2024); Shi & Jin (2025) have explored using powerful LLMs to generate verification trajectory in a CoT manner. Our OPV follows this generative paradigm but, crucially, engages human experts to provide finer-grained supervision than heuristics, aiming to overcome the inherent limitations of purely model-driven verification. Similarly, ProcessBench (Zheng et al., 2024) also engages human experts for process annotations, while we extend to a more demanding setting with more challenging queries and longer CoTs from frontier LLMs. By summarizing long CoTs into concise rationales and then verifying them, we enable larger-scale expert supervision to power our OPV.

## 6 CONCLUSION

We introduced the Outcome-based Process Verifier (OPV), which bridges outcome and process verification by operating on summarized solutions from long CoTs. Through an iterative active learning framework with expert annotations, OPV progressively improves its verification capabilities while minimizing annotation costs. Our approach achieves state-of-the-art results across multiple benchmarks, outperforming much larger models including DeepSeek-R1, despite its compact size.

---

[1]https://github.com/huggingface/Math-Verify

OPV demonstrates broad applicability throughout the reasoning pipeline: it identifies false positives in outcome-verified synthetic data, and yields consistent gains when collaborating with policy models at inference time. The accompanying OPV-BENCH dataset of 2.2k expert-annotated solutions provides a valuable resource for future research.

By enabling efficient and accurate process verification at scale, OPV addresses a critical bottleneck in developing reliable reasoning systems. As LLMs tackle increasingly complex problems with longer reasoning chains, the principle of verifying summarized rationales offers a scalable path toward more trustworthy AI-generated reasoning.

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

## THE USE OF LARGE LANGUAGE MODELS (LLMs)

We used LLMs solely for language polishing. The scientific ideas, methodology, analyses, and conclusions were entirely developed by the authors, while the LLMs assisted only in improving clarity and readability of the text.

## A ANNOTATION DETAILS

### A.1 DATA AND ANNOTATION PRELIMINARIES

**Problem Curation.** We curated math problems from widely-used benchmarks, published problem sets and open contests to ensure a broad spectrum of challenges in terms of difficulty and knowledge domains. The problems span K-12 education, high-school competitions, and undergraduate-level mathematics. To construct our dataset, we aggregated our initial data pool from three primary sources: (1) **Challenging Benchmarks (∼1k samples):** High-difficulty problems from Putnam and USAMO; (2) **Math Forums (∼10k samples):** Community-curated advanced problems from AoPS[2]; and (3) **Open-Source Datasets (∼1M samples):** Large-scale collections including NuminaMath (Li et al., 2024) and AoPS-Instruct (Mahdavi et al., 2025). We explicitly excluded multiple-choice, fill-in-the-blank, and true/false questions, as these formats allow for correct answers via "lucky guessing" or shortcuts. Consequently, we retained only open-ended calculation and proof-based problems that necessitate rigorous, step-by-step reasoning. Furthermore, to prevent data leakage, we implemented a strict decontamination pipeline against major public benchmarks (GSM8K (Cobbe et al., 2021), MATH (Hendrycks et al., 2021), OlympiadBench (He et al., 2024), Omni-MATH (Gao et al., 2024), AIME 2024, and AIME 2025). This process utilized a two-stage mechanism: first, Exact Matching via normalized string comparison to remove duplicates; and second, Semantic Matching using LLM-based embedding similarity to identify and discard queries semantically similar to the evaluation sets.

**CoT Generation and Summarization.** We sample 8 to 12 unique CoTs per problem from state-of-the-art models (R1 and Qwen families) to capture diverse reasoning paths. Initial attempts to use the default summaries following the `</think>` tag proved inadequate, as they omitted crucial intermediate steps and resisted improvement through prompt engineering. Therefore, we employ Deepseek-V3 to re-summarize the reasoning content within `<think>...</think>` tags, preserving all calculations, enumerations, and case analyses while segmenting steps uniformly with "`---`" delimiters. This procedure yields our initial unlabeled data pool, $\mathcal{D}_{\mathcal{U}} = \{(P_1, \mathcal{S}_1), \ldots, (P_m, \mathcal{S}_m)\}$, which serves as the starting point for active learning.

**Active Learning Configuration** To maximize the utility of the annotation budget, we employed a mixed sampling strategy. In each iteration loop, the data selected for expert annotation consists of two parts: (1) **Uncertainty Sampling (80%):** The majority of the budget is allocated to samples with the lowest consistency scores. We utilized a dynamic consistency threshold $\tau_t$, set to $\tau_1 = 0.25$ for the first stage and $\tau_2 = 0.5$ for the second stage, to capture increasingly subtle errors as the model improves; (2) **High-Confidence Sampling (20%):** To prevent the model from becoming overconfident or forgetting established knowledge, the remaining 20% of samples are randomly selected from the high-consistency pool. This ensures the verifier receives feedback on cases it considers "correct," allowing us to correct confident errors (false negatives) during the Expert Iteration phase.

**Annotation Protocol.** We established a precise protocol to guide expert annotation. For each sample $(P_i, \mathcal{S}_i)$, annotators provide a brief explanation $\hat{\mathcal{E}}_i$ and identify the index $\hat{\ell}_i$ of the first erroneous step. Reference solutions were provided to facilitate this process. To address ambiguity, we instructed annotators to identify "flawed but tolerable" steps—steps that are imperfect but could be easily corrected within 2-3 sentences and precede the first definitive error. Such steps were not classified as erroneous. To ensure data quality, three experts independently evaluate each solution. Annotations are valid only when: (1) all experts agree the solution is correct, or (2) at least two experts identify an error within a two-step window. This window accounts for errors that span multiple steps and resist single-step attribution. As shown in Tab. 4, annotators typically achieve stronger consensus. This protocol ensures only high-confidence labels are added to our dataset during active learning.

---

[2] https://artofproblemsolving.com/community

Table 4: Statistics of OPV-BENCH. "% $\geq n$ steps" denotes the proportion of samples whose solutions have $\geq n$ step. "% 3/3 and 2/3 agreement" denotes the proportion of samples for which each type of annotator agreement is achieved.

|  | K12 | | Highschool Competition | | Undergraduate | |
|---|---|---|---|---|---|---|
|  | **error** | **correct** | **error** | **correct** | **error** | **correct** |
| # Samples | 82 | 119 | 398 | 402 | 600 | 600 |
| # Average Steps | 6.74 | 6.70 | 7.43 | 7.11 | 6.73 | 6.71 |
| % $\geq 4$ steps | 93.9% | 97.5% | 100.0% | 100.0% | 99.0% | 97.3% |
| % $\geq 8$ steps | 30.5% | 30.3% | 40.2% | 36.1% | 29.5% | 29.3% |
| % $\geq 12$ steps | 6.1% | 5.0% | 3.5% | 2.5% | 1.3% | 3.2% |
| # Average Error Position | 3.25 | / | 2.97 | / | 2.49 | / |
| % 3/3 agreement | 76.8% | 100.0% | 85.4% | 100.0% | 100.0% | 100.0% |
| % 2/3 agreement | 23.2% | 0.0% | 14.6% | 0.0% | 0.0% | 0.0% |

## A.2 DATASET STATISTICS

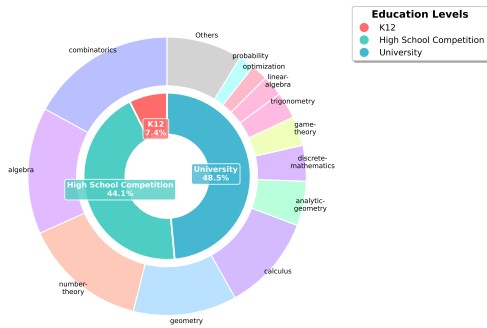

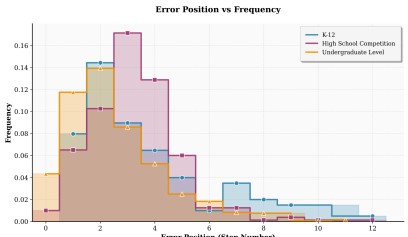

Figure 6: The topic distribution by chart of categories (inner) and domains (outer).

Figure 7: Distribution of error positions in OPV-BENCH.

The resulting OPV-BENCH comprises three subsets totaling 2,202 test cases. Tab. 4 presents detailed statistics, and Fig. 7 shows the error position distribution within erroneous samples. Since all solutions are re-summarized by Deepseek-V3 before evaluation, average step counts remain consistent across problems within each subset. However, incorrect solutions contain slightly more steps than correct ones. Most errors occur in the initial reasoning steps. High school competition problems, which require less formula application but more exploration and analysis, show a delayed error peak compared to other problems.

## B VERIFIER TRAINING DETAILS

In our experimental framework, we employ the pre-trained R1-distilled-32B DeepSeek-AI et al. (2025) model as the base architecture for fine-tuning the verifier. The verifier is trained with the following hyperparameters: 1 epoch, a learning rate of 8e-5, a sequence length of 32k, and a weight decay of 10. We utilize the DAPO training algorithm with specific configurations for the final on-policy RL stage. For the generation settings of the RL stage, we configure a global batch size of 256 and a prompt repetition factor of 8 to generate multiple samples per prompt. The optimization utilizes a global batch size of 128. Training spans 80 total steps without warmup.

## C PROMPTING DETAILS

The re-summarization prompt and verification prompt are shown in Fig. 8 and Fig.9.

As a student proficient in mathematics, you are presented with a problem and a standard solution.

Your task is to transform the given standard solution into a clear, well-structured solution that precisely articulates each reasoning step based on the material provided. Your final solution should be comprehensive, methodical, and leave no logical gaps. It should be in Chinese and should be split into steps divided by the "—" delimiter. Remember to use the same notation, symbols and language as the original solution.

If the standard solution is very brief, you may present it in just 2-3 steps. Otherwise, aim for 5-15 steps.

1. Solution Analysis: Carefully review the provided standard solution to understand the complete solution path.

2. Structured Step-by-Step Solution: Provide a thorough solution where each step follows logically from the previous one. Clearly explain every mathematical operation, theorem application, and reasoning transition mentioned in the standard solution.

3. Strict Adherence to Original Content: Your steps must strictly follow the information provided in the standard solution. Do not add new methods, concepts, or information not found in the original solution. Your task is to reorganize and clarify the existing content, not to enhance or expand it.

4. Mathematical Rigor: When presenting formulas, equations, or mathematical statements, ensure they are expressed with proper notation and justified appropriately. Define all variables and symbols when they first appear.

5. Format Requirements: Present your solution as a sequence of paragraphs, each thoroughly describing a distinct step in the solution process. Separate paragraphs with "—" as delimiters. Do not use numbered or bulleted lists. Each paragraph should represent a meaningful unit of the solution with comparable depth and detail. If the problem contains multiple sub-problems, maintain this paragraph structure throughout, treating the entire solution as a continuous sequence of steps. Conclude with a final paragraph summarizing the answer if appropriate. A sub-problem should have at least 2 paragraphs splitted. Do not use any lists or indexes nested inside a paragraph separated by "—". Since the solution is separated by the "—" delimiters that implicitly defines the index of the steps, you should not introduce any other explicit indexes, neither in numbers, English or Chinese. Every "—" delimiter should occupy a whole line.

Mathematical expressions should be clearly formatted, with proper attention to notation, subscripts, superscripts, and mathematical symbols. Use standard mathematical notation conventions.

PROBLEM STATEMENT: {problem_statement}

STANDARD SOLUTION: {standard_solution}

DETAILED SOLUTION WITH STEP-BY-STEP REASONING AND PROPER FORMAT:

Figure 8: **Prompt Template for Summarization.**

You are a mathematics and educational expert tasked with evaluating the correctness of a student's answer. The student's solution is broken down into steps, and your goal is to identify the index of the first incorrect step. The index starts at zero for the first step. If all steps are correct, you should output -1.

Instructions: - You will receive a question along with the student's answer, divided into steps. Each step is presented in a separate paragraph.

- You are encouraged to express your internal reasoning, but your final response must always include an integer within \box{STEP}. For example, if step 2 is incorrect, respond with \box{STEP2}. If all steps are correct, respond with \box{STEP-1}. Also, after you reported the incorrect step index, you should also briefly report the reason for this incorrectness.

- Some steps may initially appear incorrect but are later corrected in subsequent steps. If a reflection or revision is both accurate and reasonable, the step should be considered correct. If there are multiple reflections, consider only the final one.

- The student's answer may employ multiple approaches to solve the problem. Within a single response, some approaches may be logically sound while others may not be. If the final conclusions are correct and at least one approach is logically sound, the entire solution should be considered correct.

- In cases where the problem is ambiguous, consider all possible interpretations and determine if the student's response aligns with any of them.

- Evaluate the entire solution, as some intermediate steps might seem incorrect initially but are rectified later, such as dismissing an extraneous root. Ensure you consider the entire context and, if necessary, review the steps more than once.

- The errors to identify can be very subtle, sometimes hiding in the inexplicit applications of theorems or conditions. So you should actively checking every small logical inferences at a small granularity carefully, either in natural language or in formulas.

- To help you identify the possible errors, every first time you checking a step, you should repeat it in case you missed subtle information. Then you should check its validity by examining its logical inferences within the step/sentences/sub-sentences one by one. When a proof is required, meticulously verify the soundness of each logical step. Incomplete inductive reasoning is unacceptable. If you suspect a step is flawed, consider constructing counter-examples to test its validity.

- Every step should have solid logical basis. Guessing without proof is not allowed.

- For you convenience, you may be provided with a reference solution. The reference solution might be gaped or just a hint. It rarely is wrong but do not miss the possibility. However, some problems might ask just for one valid answer while multiple possible answers exist. In such case you should not judge the students answer is wrong because it has a different solution. Reference answers should be viewed as guides rather than absolute standards. Students may use alternative methods, notations, or approaches that are equally valid.

---

**Question**: {question}
**Answer to Verify**: {answer_to_verify}

Figure 9: **Prompt Template for Process Verification.**

# D ADDITIONAL EXPERIMENT RESULTS

Table 5: Detailed evaluation results on GSM8k partition of ProcessBench.

| Model | Absolute | | | | Approximate | | | | Rough | | | |
|---|---|---|---|---|---|---|---|---|---|---|---|---|
| | Acc | Pre | Rec | F1 | Acc | Pre | Rec | F1 | Acc | Pre | Rec | F1 |
| *With Standard Answers* | | | | | | | | | | | | |
| DeepSeek-V3-0324 | 88.5 | 81.0 | 99.5 | 89.3 | 94.5 | 90.1 | 99.5 | 94.6 | 97.8 | 96.0 | 99.5 | 97.7 |
| DeepSeek-R1-0528 | 92.0 | 86.4 | 99.0 | 92.3 | 96.0 | 93.2 | 99.0 | 96.0 | 98.5 | 97.9 | 99.0 | 98.5 |
| Qwen-Max-Preview | 91.3 | 85.0 | 99.5 | 91.6 | 95.0 | 91.0 | 99.5 | 95.0 | 98.5 | 97.5 | 99.5 | 98.5 |
| gpt-oss-120b(high) | 91.5 | 85.3 | 99.5 | 91.9 | 94.5 | 90.1 | 99.5 | 94.6 | 98.3 | 97.0 | 99.5 | 98.2 |
| Qwen2.5-Math-PRM-72B | 89.0 | 82.0 | 99.0 | 89.7 | 94.5 | 90.5 | 99.0 | 94.6 | 97.8 | 96.5 | 99.0 | 97.7 |
| DeepSeek-R1-Distill-32B | 87.8 | 81.6 | 96.4 | 88.4 | 92.5 | 89.0 | 96.4 | 92.5 | 96.0 | 95.4 | 96.4 | 95.9 |
| OPV | 91.0 | 84.6 | 99.5 | 91.4 | 94.5 | 90.1 | 99.5 | 94.6 | 98.5 | 97.5 | 99.5 | 98.5 |
| *Without Standard Answers* | | | | | | | | | | | | |
| DeepSeek-V3-0324 | 84.3 | 76.2 | 97.9 | 85.7 | 88.5 | 81.8 | 97.9 | 89.2 | 91.3 | 85.9 | 97.9 | 91.5 |
| DeepSeek-R1-0528 | 91.5 | 86.6 | 97.4 | 91.7 | 95.0 | 92.6 | 97.4 | 94.9 | 97.3 | 96.9 | 97.4 | 97.2 |
| Qwen-Max-Preview | 91.8 | 86.4 | 98.4 | 92.0 | 94.8 | 91.3 | 98.4 | 94.8 | 97.0 | 95.5 | 98.4 | 96.9 |
| gpt-oss-120b(high) | 92.3 | 87.5 | 97.9 | 92.4 | 95.8 | 93.6 | 97.9 | 95.7 | 98.0 | 97.9 | 97.9 | 97.9 |
| Qwen2.5-Math-PRM-72B | 87.3 | 79.8 | 98.5 | 88.2 | 92.0 | 86.8 | 98.5 | 92.2 | 94.5 | 90.9 | 98.5 | 94.5 |
| DeepSeek-R1-Distill-32B | 89.3 | 82.3 | 99.0 | 89.9 | 92.0 | 86.4 | 99.0 | 92.3 | 94.3 | 90.1 | 99.0 | 94.3 |
| OPV | 89.8 | 83.9 | 97.4 | 90.2 | 94.0 | 90.8 | 97.4 | 94.0 | 97.3 | 96.9 | 97.4 | 97.2 |

Table 6: Detailed evaluation results on MATH partition of ProcessBench.

| Model | Absolute | | | | Approximate | | | | Rough | | | |
|---|---|---|---|---|---|---|---|---|---|---|---|---|
| | Acc | Pre | Rec | F1 | Acc | Pre | Rec | F1 | Acc | Pre | Rec | F1 |
| *With Standard Answers* | | | | | | | | | | | | |
| DeepSeek-V3-0324 | 80.0 | 68.6 | 93.6 | 79.2 | 87.1 | 78.7 | 93.6 | 85.5 | 95.1 | 94.3 | 93.6 | 93.9 |
| DeepSeek-R1-0528 | 86.9 | 79.8 | 90.6 | 84.9 | 91.4 | 88.5 | 90.6 | 89.5 | 95.6 | 98.4 | 90.6 | 94.4 |
| Qwen-Max-Preview | 88.4 | 81.8 | 91.9 | 86.5 | 92.7 | 90.3 | 91.9 | 91.1 | 95.7 | 97.4 | 91.9 | 94.6 |
| gpt-oss-120b(high) | 88.9 | 82.6 | 92.1 | 87.1 | 93.2 | 91.2 | 92.1 | 91.7 | 96.3 | 98.7 | 92.1 | 95.3 |
| Qwen2.5-Math-PRM-72B | 83.4 | 72.6 | 95.1 | 82.3 | 89.7 | 82.3 | 95.1 | 88.2 | 95.6 | 94.2 | 95.1 | 94.6 |
| DeepSeek-R1-Distill-32B | 78.3 | 67.5 | 89.7 | 77.0 | 85.4 | 77.8 | 89.7 | 83.3 | 92.2 | 91.0 | 89.7 | 90.3 |
| OPV | 86.5 | 77.4 | 94.3 | 85.0 | 92.8 | 88.7 | 94.3 | 91.4 | 96.8 | 97.7 | 94.3 | 96.0 |
| *Without Standard Answers* | | | | | | | | | | | | |
| DeepSeek-V3-0324 | 79.8 | 68.0 | 95.1 | 79.3 | 86.5 | 77.0 | 95.1 | 85.1 | 93.3 | 89.1 | 95.1 | 92.0 |
| DeepSeek-R1-0528 | 87.7 | 79.5 | 93.8 | 86.1 | 92.3 | 88.0 | 93.8 | 90.8 | 96.5 | 97.4 | 93.8 | 95.6 |
| Qwen-Max-Preview | 89.5 | 83.1 | 93.1 | 87.8 | 93.0 | 90.0 | 93.1 | 91.5 | 96.0 | 97.4 | 93.1 | 95.2 |
| gpt-oss-120b(high) | 90.1 | 84.3 | 92.9 | 88.4 | 94.0 | 92.4 | 92.9 | 92.6 | 96.9 | 99.5 | 92.9 | 96.1 |
| Qwen2.5-Math-PRM-72B | 81.7 | 71.0 | 92.9 | 80.5 | 87.5 | 79.7 | 92.9 | 85.8 | 92.8 | 89.8 | 92.9 | 91.3 |
| DeepSeek-R1-Distill-32B | 83.7 | 72.1 | 97.8 | 83.0 | 90.7 | 82.5 | 97.8 | 89.5 | 95.1 | 90.8 | 97.8 | 94.2 |
| OPV | 87.5 | 78.6 | 95.1 | 86.1 | 93.0 | 88.5 | 95.1 | 91.7 | 96.7 | 96.7 | 95.1 | 95.9 |

Table 7: Detailed evaluation results on OlympiadBench partition of ProcessBench.

| Model | Absolute | | | | Approximate | | | | Rough | | | |
|---|---|---|---|---|---|---|---|---|---|---|---|---|
| | Acc | Pre | Rec | F1 | Acc | Pre | Rec | F1 | Acc | Pre | Rec | F1 |
| *With Standard Answers* | | | | | | | | | | | | |
| DeepSeek-V3-0324 | 73.4 | 56.7 | 91.4 | 70.0 | 82.2 | 67.6 | 91.4 | 77.7 | 93.8 | 90.6 | 91.4 | 91.0 |
| DeepSeek-R1-0528 | 81.8 | 68.1 | 87.5 | 76.6 | 89.1 | 81.7 | 87.5 | 84.5 | 95.2 | 98.3 | 87.5 | 92.6 |
| Qwen-Max-Preview | 80.7 | 66.4 | 87.2 | 75.4 | 87.6 | 78.6 | 87.2 | 82.7 | 94.6 | 96.7 | 87.2 | 91.7 |
| gpt-oss-120b(high) | 81.4 | 68.1 | 85.1 | 75.7 | 87.7 | 79.9 | 85.1 | 82.4 | 94.6 | 99.0 | 85.1 | 91.5 |
| Qwen2.5-Math-PRM-72B | 75.6 | 59.2 | 90.8 | 71.7 | 83.1 | 69.2 | 90.8 | 78.5 | 92.7 | 88.2 | 90.8 | 89.4 |
| DeepSeek-R1-Distill-32B | 65.8 | 49.8 | 80.1 | 61.4 | 75.2 | 60.2 | 80.1 | 68.7 | 88.1 | 84.1 | 80.1 | 82.0 |
| OPV | 78.1 | 62.4 | 89.6 | 73.6 | 85.3 | 73.2 | 89.6 | 80.6 | 95.5 | 97.1 | 89.6 | 93.2 |
| *Without Standard Answers* | | | | | | | | | | | | |
| DeepSeek-V3-0324 | 69.9 | 53.4 | 90.5 | 67.2 | 78.8 | 63.2 | 90.5 | 74.4 | 90.7 | 83.5 | 90.5 | 86.9 |
| DeepSeek-R1-0528 | 81.9 | 67.2 | 91.4 | 77.4 | 89.0 | 79.3 | 91.4 | 84.9 | 95.7 | 95.9 | 91.4 | 93.6 |
| Qwen-Max-Preview | 82.1 | 68.9 | 86.3 | 76.6 | 88.6 | 81.2 | 86.3 | 83.7 | 94.6 | 97.6 | 86.3 | 91.6 |
| gpt-oss-120b(high) | 82.6 | 69.6 | 86.3 | 77.1 | 88.7 | 81.4 | 86.3 | 83.8 | 95.0 | 99.0 | 86.3 | 92.2 |
| Qwen2.5-Math-PRM-72B | 74.2 | 57.9 | 88.4 | 70.0 | 81.1 | 66.7 | 88.4 | 76.1 | 89.0 | 80.9 | 88.4 | 84.5 |
| DeepSeek-R1-Distill-32B | 76.4 | 59.3 | 97.3 | 73.7 | 84.2 | 69.0 | 97.3 | 80.7 | 94.4 | 87.7 | 97.3 | 92.2 |
| OPV | 79.3 | 64.3 | 87.8 | 74.2 | 86.2 | 75.6 | 87.8 | 81.3 | 94.9 | 97.0 | 87.8 | 92.2 |

Table 8: Detailed evaluation results on Omni-MATH partition of ProcessBench.

| Model | Absolute | | | | Approximate | | | | Rough | | | |
|---|---|---|---|---|---|---|---|---|---|---|---|---|
| | Acc | Pre | Rec | F1 | Acc | Pre | Rec | F1 | Acc | Pre | Rec | F1 |
| *With Standard Answers* | | | | | | | | | | | | |
| DeepSeek-V3-0324 | 64.5 | 39.5 | 89.2 | 54.8 | 77.6 | 52.1 | 89.2 | 65.7 | 92.8 | 82.4 | 89.2 | 85.7 |
| DeepSeek-R1-0528 | 74.0 | 47.7 | 82.2 | 60.4 | 82.8 | 60.6 | 82.2 | 69.7 | 94.2 | 93.0 | 82.2 | 87.2 |
| Qwen-Max-Preview | 77.4 | 51.9 | 84.2 | 64.2 | 85.6 | 65.7 | 84.2 | 73.8 | 94.7 | 93.1 | 84.2 | 88.5 |
| gpt-oss-120b(high) | 76.9 | 51.3 | 83.4 | 63.5 | 85.8 | 66.3 | 83.4 | 73.9 | 95.4 | 97.1 | 83.4 | 89.7 |
| Qwen2.5-Math-PRM-72B | 67.8 | 42.2 | 91.3 | 57.7 | 78.1 | 52.6 | 91.3 | 66.8 | 91.0 | 76.1 | 91.3 | 83.0 |
| DeepSeek-R1-Distill-32B | 58.0 | 34.6 | 83.8 | 49.0 | 71.6 | 45.2 | 83.8 | 58.7 | 88.3 | 72.1 | 83.8 | 77.5 |
| OPV | 71.2 | 44.8 | 84.6 | 58.6 | 82.0 | 58.8 | 84.6 | 69.4 | 95.4 | 95.8 | 84.6 | 89.9 |
| *Without Standard Answers* | | | | | | | | | | | | |
| DeepSeek-V3-0324 | 61.5 | 37.4 | 88.4 | 52.5 | 74.5 | 48.4 | 88.4 | 62.6 | 90.7 | 76.6 | 88.4 | 82.1 |
| DeepSeek-R1-0528 | 76.7 | 51.0 | 88.0 | 64.5 | 84.5 | 62.7 | 88.0 | 73.2 | 95.2 | 91.8 | 88.0 | 89.8 |
| Qwen-Max-Preview | 78.6 | 53.7 | 80.9 | 64.6 | 85.7 | 66.8 | 80.9 | 73.2 | 94.2 | 94.2 | 80.9 | 87.1 |
| gpt-oss-120b(high) | 78.0 | 52.8 | 83.0 | 64.5 | 85.9 | 66.7 | 83.0 | 73.9 | 95.4 | 97.6 | 83.0 | 89.7 |
| Qwen2.5-Math-PRM-72B | 69.9 | 43.8 | 88.4 | 58.6 | 79.7 | 54.9 | 88.4 | 67.7 | 90.4 | 75.8 | 88.4 | 81.6 |
| DeepSeek-R1-Distill-32B | 69.0 | 43.6 | 97.1 | 60.2 | 78.9 | 53.4 | 97.1 | 68.9 | 92.9 | 78.5 | 97.1 | 86.8 |
| OPV | 72.5 | 46.1 | 83.0 | 59.3 | 82.7 | 60.2 | 83.0 | 69.8 | 95.2 | 96.6 | 83.0 | 89.3 |

Table 9: Detailed evaluation results on K-12 partition of OPV-Bench.

| Model | Absolute | | | | Approximate | | | | Rough | | | |
|---|---|---|---|---|---|---|---|---|---|---|---|---|
| | Acc | Pre | Rec | F1 | Acc | Pre | Rec | F1 | Acc | Pre | Rec | F1 |
| *With Standard Answers* | | | | | | | | | | | | |
| DeepSeek-V3-0324 | 75.7 | 80.3 | 78.3 | 79.3 | 77.7 | 83.2 | 78.3 | 80.7 | 81.2 | 88.7 | 78.3 | 83.2 |
| DeepSeek-R1-0528 | 86.6 | 85.4 | 93.3 | 89.2 | 87.1 | 86.1 | 93.3 | 89.5 | 88.6 | 88.1 | 93.3 | 90.6 |
| Qwen-Max-Preview | 87.6 | 87.9 | 91.6 | 89.7 | 88.6 | 89.3 | 91.6 | 90.5 | 90.0 | 91.6 | 91.6 | 91.6 |
| gpt-oss-120b(high) | 82.1 | 79.9 | 93.3 | 86.1 | 84.6 | 82.8 | 93.3 | 87.8 | 87.6 | 86.7 | 93.3 | 89.9 |
| Qwen2.5-Math-PRM-72B | 60.4 | 61.4 | 90.0 | 73.0 | 61.4 | 62.1 | 90.0 | 73.5 | 68.8 | 67.9 | 90.0 | 77.4 |
| DeepSeek-R1-Distill-32B | 75.1 | 74.5 | 88.2 | 80.8 | 77.1 | 76.6 | 88.2 | 82.0 | 80.6 | 80.8 | 88.2 | 84.3 |
| OPV-Stage1 | 76.6 | 72.2 | 98.3 | 83.3 | 78.6 | 74.1 | 98.3 | 84.5 | 81.1 | 76.5 | 98.3 | 86.0 |
| OPV-Stage2 | 86.6 | 82.4 | 98.3 | 89.7 | 87.6 | 83.6 | 98.3 | 90.4 | 90.5 | 87.3 | 98.3 | 92.5 |
| OPV-Stage3-w/o RL | 87.6 | 88.5 | 90.8 | 89.6 | 88.6 | 90.0 | 90.8 | 90.4 | 90.0 | 92.3 | 90.8 | 91.5 |
| OPV | 86.1 | 89.6 | 86.6 | 88.0 | 87.1 | 91.2 | 86.6 | 88.8 | 88.1 | 92.8 | 86.6 | 89.6 |
| *Without Standard Answers* | | | | | | | | | | | | |
| DeepSeek-V3-0324 | 70.3 | 71.7 | 82.5 | 76.7 | 71.8 | 73.3 | 82.5 | 77.7 | 76.2 | 78.6 | 82.5 | 80.5 |
| DeepSeek-R1-0528 | 83.1 | 80.1 | 95.0 | 86.9 | 83.1 | 80.1 | 95.0 | 86.9 | 85.1 | 82.5 | 95.0 | 88.3 |
| Qwen-Max-Preview | 84.6 | 83.3 | 92.4 | 87.7 | 86.1 | 85.3 | 92.4 | 88.7 | 87.6 | 87.3 | 92.4 | 89.8 |
| gpt-oss-120b(high) | 79.0 | 77.7 | 90.8 | 83.7 | 81.0 | 80.0 | 90.8 | 85.0 | 85.0 | 85.0 | 90.8 | 87.8 |
| Qwen2.5-Math-PRM-72B | 61.9 | 62.9 | 87.5 | 73.2 | 62.4 | 63.3 | 87.5 | 73.4 | 64.4 | 64.8 | 87.5 | 74.5 |
| DeepSeek-R1-Distill-32B | 73.6 | 70.6 | 95.0 | 81.0 | 74.6 | 71.5 | 95.0 | 81.6 | 77.1 | 73.9 | 95.0 | 83.1 |
| OPV-Stage1 | 72.6 | 69.3 | 96.6 | 80.7 | 73.1 | 69.7 | 96.6 | 81.0 | 74.6 | 71.0 | 96.6 | 81.9 |
| OPV-Stage2 | 83.1 | 80.1 | 95.0 | 86.9 | 83.6 | 80.7 | 95.0 | 87.3 | 87.1 | 85.0 | 95.0 | 89.7 |
| OPV-Stage3-w/o RL | 84.6 | 84.4 | 90.8 | 87.5 | 85.1 | 85.0 | 90.8 | 87.8 | 87.1 | 87.8 | 90.8 | 89.3 |
| OPV | 87.6 | 86.7 | 93.3 | 89.9 | 88.1 | 87.4 | 93.3 | 90.2 | 90.5 | 91.0 | 93.3 | 92.1 |

Table 10: Detailed evaluation results on High School Competition partition of OPV-Bench.

| Model | Absolute | | | | Approximate | | | | Rough | | | |
|---|---|---|---|---|---|---|---|---|---|---|---|---|
| | Acc | Pre | Rec | F1 | Acc | Pre | Rec | F1 | Acc | Pre | Rec | F1 |
| *With Standard Answers* | | | | | | | | | | | | |
| DeepSeek-V3-0324 | 72.3 | 64.6 | 88.1 | 74.6 | 78.4 | 71.6 | 88.1 | 79.0 | 85.3 | 81.5 | 88.1 | 84.7 |
| DeepSeek-R1-0528 | 76.0 | 69.4 | 93.5 | 79.7 | 82.3 | 76.4 | 93.5 | 84.1 | 88.9 | 85.7 | 93.5 | 89.4 |
| Qwen-Max-Preview | 73.4 | 67.5 | 90.8 | 77.4 | 79.4 | 74.0 | 90.8 | 81.6 | 87.3 | 84.9 | 90.8 | 87.7 |
| gpt-oss-120b(high) | 68.9 | 64.2 | 86.1 | 73.5 | 78.6 | 75.1 | 86.1 | 80.2 | 86.3 | 86.5 | 86.1 | 86.3 |
| Qwen2.5-Math-PRM-72B | 49.5 | 47.0 | 97.6 | 63.4 | 51.3 | 47.9 | 97.6 | 64.3 | 56.6 | 50.9 | 97.6 | 66.9 |
| DeepSeek-R1-Distill-32B | 73.4 | 66.0 | 97.0 | 78.6 | 79.1 | 71.6 | 97.0 | 82.4 | 83.1 | 76.0 | 97.0 | 85.3 |
| OPV-Stage1 | 71.8 | 64.5 | 97.5 | 77.6 | 76.3 | 68.5 | 97.5 | 80.5 | 80.1 | 72.5 | 97.5 | 83.1 |
| OPV-Stage2 | 75.8 | 69.0 | 94.0 | 79.6 | 80.8 | 74.4 | 94.0 | 83.1 | 87.0 | 82.5 | 94.0 | 87.9 |
| OPV-Stage3-w/o RL | 79.9 | 78.6 | 82.3 | 80.4 | 85.5 | 88.0 | 82.3 | 85.1 | 90.3 | 97.9 | 82.3 | 89.5 |
| OPV | 83.0 | 81.7 | 85.3 | 83.5 | 87.9 | 90.0 | 85.3 | 87.6 | 91.3 | 96.9 | 85.3 | 90.7 |
| *Without Standard Answers* | | | | | | | | | | | | |
| DeepSeek-V3-0324 | 66.5 | 58.6 | 90.7 | 71.2 | 73.1 | 64.6 | 90.7 | 75.4 | 81.2 | 74.0 | 90.7 | 81.5 |
| DeepSeek-R1-0528 | 67.0 | 61.4 | 92.8 | 73.9 | 74.5 | 68.1 | 92.8 | 78.5 | 83.3 | 78.0 | 92.8 | 84.8 |
| Qwen-Max-Preview | 69.4 | 63.8 | 90.3 | 74.8 | 77.6 | 72.2 | 90.3 | 80.2 | 86.4 | 83.8 | 90.3 | 87.0 |
| gpt-oss-120b(high) | 67.1 | 62.7 | 85.3 | 72.3 | 76.9 | 73.1 | 85.3 | 78.8 | 84.9 | 84.7 | 85.3 | 85.0 |
| Qwen2.5-Math-PRM-72B | 49.5 | 47.0 | 97.3 | 63.4 | 51.1 | 47.8 | 97.3 | 64.1 | 54.4 | 49.6 | 97.3 | 65.7 |
| DeepSeek-R1-Distill-32B | 64.6 | 59.0 | 97.5 | 73.5 | 68.4 | 61.7 | 97.5 | 75.6 | 73.1 | 65.7 | 97.5 | 78.5 |
| OPV-Stage1 | 65.8 | 59.9 | 96.3 | 73.9 | 70.8 | 63.9 | 96.3 | 76.8 | 76.1 | 68.7 | 96.3 | 80.2 |
| OPV-Stage2 | 69.8 | 63.6 | 93.3 | 75.6 | 77.8 | 71.3 | 93.3 | 80.8 | 85.4 | 80.7 | 93.3 | 86.5 |
| OPV-Stage3-w/o RL | 76.4 | 72.1 | 86.3 | 78.6 | 82.4 | 80.1 | 86.3 | 83.1 | 89.0 | 91.3 | 86.3 | 88.8 |
| OPV | 78.3 | 72.6 | 91.0 | 80.8 | 85.4 | 81.9 | 91.0 | 86.2 | 90.8 | 90.6 | 91.0 | 90.8 |

Table 11: Detailed evaluation results on Undergraduate partition of OPV-Bench.

| Model | Absolute | | | | Approximate | | | | Rough | | | |
|---|---|---|---|---|---|---|---|---|---|---|---|---|
| | Acc | Pre | Rec | F1 | Acc | Pre | Rec | F1 | Acc | Pre | Rec | F1 |
| *With Standard Answers* | | | | | | | | | | | | |
| DeepSeek-V3-0324 | 62.2 | 59.1 | 78.8 | 67.5 | 66.6 | 63.3 | 78.8 | 70.2 | 71.5 | 68.7 | 78.8 | 73.4 |
| DeepSeek-R1-0528 | 58.3 | 57.0 | 68.9 | 62.4 | 63.6 | 62.5 | 68.9 | 65.5 | 69.9 | 70.5 | 68.9 | 69.7 |
| Qwen-Max-Preview | 58.7 | 56.9 | 73.3 | 64.1 | 64.2 | 62.1 | 73.3 | 67.3 | 70.0 | 68.9 | 73.3 | 71.1 |
| gpt-oss-120b(high) | 52.8 | 52.4 | 67.4 | 58.9 | 62.0 | 61.1 | 67.4 | 64.1 | 67.0 | 72.5 | 67.4 | 69.8 |
| Qwen2.5-Math-PRM-72B | 56.2 | 53.5 | 95.2 | 68.5 | 57.6 | 54.3 | 95.2 | 69.2 | 62.0 | 57.2 | 95.2 | 71.5 |
| DeepSeek-R1-Distill-32B | 67.9 | 62.3 | 91.5 | 74.1 | 71.7 | 65.7 | 91.5 | 76.5 | 75.4 | 69.4 | 91.5 | 78.9 |
| OPV-Stage1 | 65.8 | 60.3 | 93.3 | 73.3 | 69.4 | 63.2 | 93.3 | 75.4 | 72.7 | 66.2 | 93.3 | 77.5 |
| OPV-Stage2 | 66.5 | 62.7 | 82.2 | 71.1 | 71.4 | 67.8 | 82.2 | 74.3 | 75.7 | 72.9 | 82.2 | 77.2 |
| OPV-Stage3-w/o RL | 72.2 | 74.2 | 68.6 | 71.3 | 76.2 | 81.0 | 68.6 | 74.3 | 81.0 | 91.5 | 68.6 | 78.4 |
| OPV | 75.1 | 76.6 | 72.5 | 74.5 | 79.6 | 84.7 | 72.5 | 78.1 | 84.3 | 95.1 | 72.5 | 82.3 |
| *Without Standard Answers* | | | | | | | | | | | | |
| DeepSeek-V3-0324 | 53.3 | 52.2 | 79.2 | 62.9 | 57.9 | 55.5 | 79.2 | 65.3 | 63.9 | 60.6 | 79.2 | 68.7 |
| DeepSeek-R1-0528 | 46.4 | 47.5 | 64.3 | 54.7 | 52.9 | 52.5 | 64.3 | 57.8 | 62.3 | 62.0 | 64.3 | 63.1 |
| Qwen-Max-Preview | 51.9 | 51.6 | 68.3 | 58.8 | 57.1 | 56.0 | 68.3 | 61.5 | 65.8 | 65.3 | 68.3 | 66.7 |
| gpt-oss-120b(high) | 48.9 | 49.3 | 63.0 | 55.3 | 58.2 | 57.7 | 63.0 | 60.2 | 68.2 | 70.5 | 63.0 | 66.5 |
| Qwen2.5-Math-PRM-72B | 55.2 | 52.9 | 94.0 | 67.7 | 57.1 | 54.1 | 94.0 | 68.7 | 60.4 | 56.2 | 94.0 | 70.4 |
| DeepSeek-R1-Distill-32B | 58.0 | 55.1 | 88.9 | 68.0 | 61.3 | 57.4 | 88.9 | 69.8 | 65.7 | 60.9 | 88.9 | 72.3 |
| OPV-Stage1 | 56.0 | 53.8 | 86.3 | 66.3 | 59.7 | 56.4 | 86.3 | 68.2 | 64.9 | 60.6 | 86.3 | 71.2 |
| OPV-Stage2 | 58.2 | 56.4 | 73.7 | 63.9 | 63.2 | 61.1 | 73.7 | 66.8 | 71.2 | 70.3 | 73.7 | 71.9 |
| OPV-Stage3-w/o RL | 61.4 | 60.8 | 65.1 | 62.9 | 67.4 | 68.4 | 65.1 | 66.7 | 73.1 | 77.6 | 65.1 | 70.8 |
| OPV | 65.4 | 63.7 | 72.5 | 67.8 | 71.9 | 71.8 | 72.5 | 72.2 | 77.0 | 79.8 | 72.5 | 76.0 |

# E  CASE STUDIES

Here we list interesting cases containing process errors that were found by OPV across different datasets.

**Question:**
Let $a, b, c > 0$ such that $a + b \leq c$. Find the minimum value of $P = \left(a^4 + b^4 + c^4\right)\left(\frac{1}{a^4} + \frac{1}{b^4} + \frac{1}{c^4}\right)$.

**Model Answer from AM-DeepSeek-R1-0528-Distilled:**
Given positive real numbers $a$, $b$, and $c$ with the constraint $a + b \leq c$, the goal is to minimize the expression

$$P = \left(a^4 + b^4 + c^4\right)\left(\frac{1}{a^4} + \frac{1}{b^4} + \frac{1}{c^4}\right).$$

The expression $P$ is symmetric in $a$ and $b$, and the constraint is also symmetric in $a$ and $b$. Therefore, the minimum likely occurs when $a = b$. Assuming $a = b$, the constraint becomes $2a \leq c$. Let $t = c/a$, so $t \geq 2$. Then,

$$P = \left(a^4 + a^4 + (ta)^4\right)\left(\frac{1}{a^4} + \frac{1}{a^4} + \frac{1}{(ta)^4}\right) = (2a^4 + t^4 a^4)\left(\frac{2}{a^4} + \frac{1}{t^4 a^4}\right).$$

The $a^4$ terms cancel, giving

$$P = (2 + t^4)\left(2 + \frac{1}{t^4}\right) = 4 + 2t^4 + \frac{2}{t^4} + 1 = 5 + 2t^4 + \frac{2}{t^4}.$$

Substituting $u = t^4$ (so $u \geq 16$), $P = 5 + 2u + \frac{2}{u}$. The derivative is $P'(u) = 2 - \frac{2}{u^2}$, which is positive for $u > 1$. Since $u \geq 16 > 1$, $P(u)$ is increasing for $u \geq 16$, and the minimum occurs at $u = 16$ (i.e., $t = 2$). At this point,

$$P = 5 + 2(16) + \frac{2}{16} = 5 + 32 + 0.125 = 37.125 = \frac{297}{8}.$$

This value is achieved when $a = b$ and $c = 2a$, satisfying $a + b = c$ (equality in the constraint).
To confirm this is the global minimum, consider cases where $a \neq b$ or $a + b < c$. When $a + b < c$, setting $a = b = 1$ and $c > 2$ gives $P > 297/8$. For $a \neq b$ with $a + b = c$, define $r = b/a > 0$, so

$$P = \left[1 + r^4 + (1 + r)^4\right]\left[1 + \frac{1}{r^4} + \frac{1}{(1+r)^4}\right].$$

At $r = 1$, $P = 297/8$. The derivative at $r = 1$ is zero, and $P(r) > 297/8$ for $r \neq 1$. As $r \to 0^+$ or $r \to \infty$, $P \to \infty$. Thus, under $a + b = c$, the minimum is $297/8$. When $a + b < c$, $P > 297/8$, so the minimum is achieved when equality holds in the constraint.
Thus, the minimum value of $P$ is $\frac{297}{8}$.

**Judgments of OPV:**
The first error occurs in Step 1 where the student assumes $a = b$ based solely on symmetry without rigorous justification. While the problem is symmetric in $a$ and $b$, the minimum might not necessarily occur at $a = b$ without proof. Although the student later verifies this assumption through additional analysis, the initial step lacks a formal argument (e.g., using inequalities, convexity, or perturbation methods) to confirm that deviations from $a = b$ would increase $P$. This makes Step 1 logically incomplete and thus incorrect.

Figure 10: **A case study sampled from AM-DeepSeek-R1-0528-Distilled dataset where the reasoning process is defective despite the final answer being correct.** The rigor of the inequality is not proven in the model's proposed answer, which is identified by OPV. In the generations, we use ref color to present the defective content.

