# OpenReview forum: "OPV: Outcome-based Process Verifier for Efficient Long Chain-of-Thought Verification"
_ICLR.cc/2026/Conference — Submitted to ICLR 2026_

### Official Review · Reviewer_CTy4 · 2025-10-28

**Soundness:** 3
**Presentation:** 3
**Contribution:** 3
**Rating:** 6
**Confidence:** 4

**Summary:**

The paper proposes an OPV for long-form reasoning. Instead of scoring raw, lengthy CoT traces directly, the method first compresses them into concise, linear “summaries” and then predicts whether each summarized step is correct, aiming to localize the first erroneous step and provide a short explanation. Training combines RFT-style supervision with an active learning loop that queries experts for the most uncertain cases and further improves the verifier. The authors position this as a practical route to reliable process supervision, claiming it scales better than prior PVs that operate on full CoTs and can be used for dataset cleaning and improved test-time selection.

**Strengths:**

- **Motivation & intuition for process supervision:** The paper tackles the central challenge of reliable process supervision. Using summaries as a proxy for long CoTs is intuitively appealing: it can reduce noise, strip incidental verbosity, and make step-level correctness more predictable.
- **Practical training framework:** The active learning setup is a sensible pairing with the proposed verifier—both for label efficiency and for turning process supervision into a usable training pipeline rather than a one-off annotation exercise.

**Weaknesses:**

- **Missing PV baselines:** The evaluation omits a set of established process reward models (PVs) as baselines. Including them would allow clearer comparisons and help assess whether the gains come from summarization, the active learning loop, or other factors. It is unclear what prevented their inclusion.
- **Uncertainty measurement in active learning:** The uncertainty measure appears to mix aleatoric and epistemic components, which can misguide acquisition. I assume active learning aims to learn from the samples with high epistemic uncertainty?

**Questions:**

- **On baselines (W1):** Which prior PVs were considered and why were they excluded? Could the authors report results for representative PVs under a matched training budget?
- **On uncertainty (W2):** How, if at all, do the authors separate or control for aleatoric vs. epistemic uncertainty in their selection criterion? Have they tested alternative acquisition functions (e.g., variance based approach)?

---

> ### Author Response · Authors · 2025-12-03
> **Response to Weakness1/Question1 about Missing Baselines**
>
> > Missing PV baselines: The evaluation omits a set of established process reward models (PVs) as baselines. Including them would allow clearer comparisons and help assess whether the gains come from summarization, the active learning loop, or other factors. It is unclear what prevented their inclusion.
> >
> > On baselines (W1): Which prior PVs were considered and why were they excluded? Could the authors report results for representative PVs under a matched training budget?
>
> We agree with the reviewer that including established Process Reward Models (PRMs) provides crucial context for evaluating our method.
>
> **Reason for Initial Exclusion.** Our initial focus was on Long-Reasoning Models (LRMs) because the landscape of reasoning has shifted significantly. Traditional PRMs are typically discriminative models trained on standard instruction-tuned backbones. In our preliminary experiments, we observed that these dedicated PRMs often underperform compared to recent LRMs (like DeepSeek-R1) utilizing zero-shot prompting. The generative long chain-of-thought (CoT) capability of LRMs significantly enhances their verification ability, effectively making them the new state-of-the-art baseline.
>
> **Inclusion of Qwen2.5-Math-PRM-72B.** To address your concern and provide a clearer comparison, we have added Qwen2.5-Math-PRM-72B to our benchmark evaluation (see updated Table 1; an excerpt is provided below). This model represents a strong, established baseline: it is a discriminative PRM trained on labels that integrate Monte Carlo estimation and LLM-as-judge.
>
> The results confirm our hypothesis and highlight the advantages of our approach: Even our base model, R1-Distill-Qwen-32B (an LRM), consistently outperforms the larger Qwen2.5-Math-PRM-72B across benchmarks. OPV-32B further widens this gap. The significant performance margin of OPV over Qwen2.5-Math-PRM-72B suggests that the high-quality, expert-annotated data acquired through our human-in-the-loop pipeline provides far superior training signals than the heuristic labels used to train traditional PRMs.
>
> **Table A: ProcessBench (Without Standard Answers)**
>
> | Model | Precise/Abs. Accuracy | Precise/Abs. F1 | Approximate Accuracy | Approximate F1 | Rough Accuracy | Rough F1 |
> | :--- | :--- | :--- | :--- | :--- | :--- | :--- |
> | Qwen2.5-Math-PRM-72B | 76.7 | 73.2 | 83.9 | 79.8 | 91.2 | 87.8 |
> | R1-Distill-Qwen-32B | 75.8 | 73.6 | 83.6 | 80.5 | 93.3 | 91.1 |
> | OPV-32B | 80.9 | 76.8 | 88.1 | 84.1 | 95.8 | 93.8 |
>
> **Table B: OPV-Bench (Without Standard Answers)**
>
> | Model | Precise/Abs. Accuracy | Precise/Abs. F1 | Approximate Accuracy | Approximate F1 | Rough Accuracy | Rough F1 |
> | :--- | :--- | :--- | :--- | :--- | :--- | :--- |
> | Qwen2.5-Math-PRM-72B | 55.1 | 66.0 | 58.2 | 67.6 | 63.4 | 70.4 |
> | R1-Distill-Qwen-32B | 61.7 | 71.1 | 65.0 | 72.9 | 69.4 | 75.5 |
> | OPV-32B | 71.9 | 74.7 | 78.2 | 79.1 | 83.1 | 83.1 |

---

> ### Author Response · Authors · 2025-12-03
> **Response to Weakness2/Question2 about Uncertainty Measurement**
>
> > Uncertainty measurement in active learning: The uncertainty measure appears to mix aleatoric and epistemic components, which can misguide acquisition. I assume active learning aims to learn from the samples with high epistemic uncertainty?
> >
> > On uncertainty (W2): How, if at all, do the authors separate or control for aleatoric vs. epistemic uncertainty in their selection criterion? Have they tested alternative acquisition functions (e.g., variance based approach)?
>
> We appreciate the reviewer's theoretical rigor regarding the distinction between aleatoric and epistemic uncertainty. We argue that in our specific domain of mathematical process verification, the uncertainty captured by our self-consistency metric is predominantly epistemic, making it a highly effective acquisition function for active learning.
>
> 1. **Minimal Aleatoric Uncertainty in Mathematical Reasoning**. Aleatoric uncertainty typically arises from inherent noise or ambiguity in the data labels (e.g., blurry images or subjective sentiment). In contrast, mathematical reasoning is strictly logical and deterministic. A reasoning step is objectively either logically sound or fallacious based on its premises; there are no "ambiguous labels" or inherent randomness in the ground truth correctness of a mathematical derivation. Therefore, the "data noise" component (aleatoric uncertainty) is negligible in this context.
>
> 2. **Variance Reflects Epistemic Uncertainty**. Since the ground truth is deterministic, any variance in the verifier's predictions stems from the model's own lack of knowledge or reasoning capability, not from the data itself. Our OPV employs a generative verification approach: for each sample, it generates a complete chain of thought before outputting a verdict and error location. If the model produces divergent verdicts across multiple sampling runs (e.g., flagging Step 3 in one run but Step 5 in another), this disagreement directly indicates the model's internal confusion and inability to robustly discern the logic—precisely the definition of epistemic uncertainty.
>
> Thus, in the deterministic setting of mathematics, the consistency of the verifier's outputs serves as a direct and robust proxy for epistemic uncertainty, allowing us to effectively target the samples where the model is most "confused" for expert annotation.

---

### Official Review · Reviewer_k3ry · 2025-10-29

**Soundness:** 2
**Presentation:** 3
**Contribution:** 2
**Rating:** 4
**Confidence:** 2

**Summary:**

This paper introduces an outcome based process verifier (OPV), aiming to bridge in between the process reward model and the outcome reward model. The framework uses iterative data refinement: human experts improve process annotations each round, and the model is retrained by using both human-labeled and model-generated data. Through this loop, OPV enhances both model performance and annotation quality over time.

**Strengths:**

+ Existing PRM approaches use Monte Carlo rollouts which are noisy. Different from previous automated rollouts, this paper utilizes humans in the loop to improve the data and model quality iteratively.

+ Curate and iterate to create the OPV-Bench dataset, which could be valuable for process verification methods.

**Weaknesses:**

- Missing model comparison. This work is positioned in-between the ORM and PRM, but does not provide direct comparisons with existing methods (e.g., Qwen PRM or other ORM baselines). For the benchmark it would be important to understand 1. How do the other models compare on the process bench 2? How do the other models compare with the OPV-BENCH?

- Missing Dataset construction details. Authors only mentioned problem curation as K-12 education, high-school competitions, and undergraduate-level mathematics in A.1. There should be more specific sources on the problem curation.

- Since OPV is trained on the dataset with similar distribution as OPV-Bench, performance gains may reflect dataset alignment. The model needs to be evaluated on more datasets other than Process Bench, such as full GSM8K, MATH, Minerva, or MMLU, etc. In addition,  Process Bench should have subcategory results of the GSM8K, MATH, Olympiad, and Omni-MATH.

- The pipeline is rather complex. It requires multiple stages for training, and for each training it has expert iteration then online RL. What would the total training cost with this approach? It is important to compare this with Monte Carlo rollout methods. Also it would be important to understand the human annotation cost.

**Questions:**

(1) L190-209: how does OPV handle cases where the model exhibits high confidence yet produces consistent but incorrect reasoning, which is a common issue in RL-trained models?

(2) Best-of-N ranking (Section 4.2, L402): Could the authors clarify how ranking is determined? Is OPV used directly to score and rank the outputs through the index of the first incorrect answer?

---

> ### Author Response · Authors · 2025-12-03
> **Response to Weakness1 about Missing Model Comparison**
>
> > Missing model comparison. This work is positioned in-between the ORM and PRM, but does not provide direct comparisons with existing methods (e.g., Qwen PRM or other ORM baselines). For the benchmark it would be important to understand 1. How do the other models compare on the process bench 2? How do the other models compare with the OPV-BENCH?
>
> We agree with the reviewer that including existing models provides crucial context for evaluating our method.
>
> **Reason for Initial Exclusion**. Our initial focus was on Long-Reasoning Models (LRMs) because the landscape of reasoning has shifted significantly. Traditional PRMs are typically discriminative models trained on standard instruction-tuned backbones. In our preliminary experiments, we observed that these dedicated PRMs often underperform compared to recent LRMs (like DeepSeek-R1) utilizing zero-shot prompting. The generative long chain-of-thought (CoT) capability of LRMs significantly enhances their verification ability, effectively making them the new state-of-the-art baseline.
>
> Regarding ORMs, they are intrinsically incapable of step-level verification on benchmarks like ProcessBench, as they only assess final answer correctness against a given ground truth. Our analysis reveals that 30.6% of solutions in ProcessBench and 32.0% in OPV-Bench are "false positives" (correct final answer but flawed reasoning). ORMs inherently fail to detect these cases, rendering them unsuitable for this specific evaluation.
>
> **Inclusion of Qwen2.5-Math-PRM-72B**. To address your concern and provide a clearer comparison, we have added Qwen2.5-Math-PRM-72B to our benchmark evaluation (see updated Table 1; an excerpt is provided below). This model represents a strong, established baseline: it is a discriminative PRM trained on labels that integrate Monte Carlo estimation and LLM-as-judge.
>
> The results confirm our hypothesis and highlight the advantages of our approach: Even our base model, R1-Distill-Qwen-32B (an LRM), consistently outperforms the larger Qwen2.5-Math-PRM-72B across benchmarks. OPV-32B further widens this gap. The significant performance margin of OPV over Qwen2.5-Math-PRM-72B suggests that the high-quality, expert-annotated data acquired through our human-in-the-loop pipeline provides far superior training signals than the heuristic labels used to train traditional PRMs.
>
> **Table A: ProcessBench (Without Standard Answers)**
>
> | Model | Precise/Abs. Accuracy | Precise/Abs. F1 | Approximate Accuracy | Approximate F1 | Rough Accuracy | Rough F1 |
> | :--- | :--- | :--- | :--- | :--- | :--- | :--- |
> | Qwen2.5-Math-PRM-72B | 76.7 | 73.2 | 83.9 | 79.8 | 91.2 | 87.8 |
> | R1-Distill-Qwen-32B | 75.8 | 73.6 | 83.6 | 80.5 | 93.3 | 91.1 |
> | OPV-32B | 80.9 | 76.8 | 88.1 | 84.1 | 95.8 | 93.8 |
>
> **Table B: OPV-Bench (Without Standard Answers)**
>
> | Model | Precise/Abs. Accuracy | Precise/Abs. F1 | Approximate Accuracy | Approximate F1 | Rough Accuracy | Rough F1 |
> | :--- | :--- | :--- | :--- | :--- | :--- | :--- |
> | Qwen2.5-Math-PRM-72B | 55.1 | 66.0 | 58.2 | 67.6 | 63.4 | 70.4 |
> | R1-Distill-Qwen-32B | 61.7 | 71.1 | 65.0 | 72.9 | 69.4 | 75.5 |
> | OPV-32B | 71.9 | 74.7 | 78.2 | 79.1 | 83.1 | 83.1 |

---

> ### Author Response · Authors · 2025-12-03
> **Response to Weakness2 about Missing Dataset construction details**
>
> > Missing Dataset construction details. Authors only mentioned problem curation as K-12 education, high-school competitions, and undergraduate-level mathematics in A.1. There should be more specific sources on the problem curation.
>
> We appreciate the reviewer's scrutiny regarding data hygiene. We assure you that OPV-BENCH was constructed with a rigorous decontamination pipeline to prevent data leakage. (We have also supplemented Appendix A.1 with the following details.)
>
> 1. **Data Sources and Filtering** Our initial data pool was aggregated from three primary sources:
>   - Challenging Benchmarks (~1k): High-difficulty problems from Putnam and USAMO.
>   - Math Forums (~10k): Community-curated advanced problems from AoPS.
>   - Open-Source Datasets (~1000k): Large-scale collections including Numina-Math and AoPS-Instruct.
>
> 2. **Filtering Criteria**: First we remove multi-modal queries (i.e., those involving images). Then we explicitly excluded multiple-choice, fill-in-the-blank, and True/False questions, as correct answers for these can often be obtained via "lucky guessing" or shortcuts. We retained only open-ended calculation and proof-based problems that necessitate rigorous, step-by-step reasoning.
>
> 3. **Rigorous Decontamination Pipeline** To prevent data leakage, we implemented a strict decontamination process against major public benchmarks (GSM8K, MATH, OlympiadBench, Omni-MATH, AIME 2024, and AIME 2025). We employed a two-stage filtering mechanism:
>   - **Exact Matching**: Normalized string matching to remove duplicates.
>   - **Semantic Matching**: LLM-based embedding similarity to identify and remove queries that were semantically similar to those benchmarks.

---

> ### Author Response · Authors · 2025-12-03
> **Response to Weakness3 about Broader Evaluation**
>
> > Since OPV is trained on the dataset with similar distribution as OPV-Bench, performance gains may reflect dataset alignment. The model needs to be evaluated on more datasets other than Process Bench, such as full GSM8K, MATH, Minerva, or MMLU, etc. In addition, Process Bench should have subcategory results of the GSM8K, MATH, Olympiad, and Omni-MATH.
>
> We appreciate the reviewer's rigorous inquiry regarding dataset alignment and the request for broader evaluation.
>
> 1. We would like to clarify that benchmarks such as GSM8K, MATH, Minerva, and MMLU are designed to evaluate policy models, not process verifiers. Currently, ProcessBench is the primary open-source benchmark specifically annotated with granular step-level labels required to evaluate a verifier's precision. Therefore, we cannot directly evaluate the verifier's error-detection accuracy on benchmarks without existing process annotations.
>
> 2. We respectfully disagree that our performance gains are merely due to dataset alignment.
>
> - Out-of-Distribution Performance: While OPV-Bench aligns with our training distribution, ProcessBench does not. As shown in Table 1, our OPV-32B achieves consistent and significant gains on ProcessBench (Rough F1: 93.8 vs. 91.1 for the base model). This demonstrates that the verification capabilities learned via our expert iteration pipeline generalize effectively to out-of-distribution reasoning trajectories.
>
> - Downstream Robustness: Furthermore, the model's robustness is validated in practical applications beyond static benchmarks. As detailed in Section 4, OPV successfully provides fine-grained supervision for training and enhances test-time scaling through collaborative reasoning. These successful downstream applications confirm that the model has learned robust verification logic, not just dataset-specific artifacts.
> 3. We have included detailed subcategory results for both ProcessBench (GSM8K, MATH, Olympiad, Omni-MATH) and OPV-Bench (K-12, High School Competition, Undergraduate) in Appendix D. These results reveal a compelling trend: OPV's performance advantage widens as problem difficulty increases.
> - On the OPV-Bench (K-12) split, OPV outperforms the powerful gpt-oss-120b by +7.1% in F1 score.
> - On the significantly harder OPV-Bench (Undergraduate) split, this gain expands to +12.5%

---

> ### Author Response · Authors · 2025-12-03
> **Response to Weakness4 about Total Training Cost**
>
> > The pipeline is rather complex. It requires multiple stages for training, and for each training it has expert iteration then online RL. What would the total training cost with this approach? It is important to compare this with Monte Carlo rollout methods. Also it would be important to understand the human annotation cost.
>
> Total compute for our 3-stage pipeline is approx. 6.7k GPU-hours (SFT: 64 GPUs/4-10h per stage; Expert Iteration: 128 GPUs/12h per stage; Online RL: 128 GPUs/6h total). Regarding annotation, we collected >180k annotations (3-vote scheme) to yield 40k valid solutions. While computationally higher than Monte Carlo rollouts, this cost is critical for preventing reward hacking. Monte Carlo methods rely on proxy rewards that can be exploited; our human-in-the-loop annotations provide the ground-truth signals necessary to penalize such exploitation, ensuring model robustness that rollout methods alone cannot achieve.

---

> ### Author Response · Authors · 2025-12-03
> **Response to Question1 about Confident Errors**
>
> > (1) L190-209: how does OPV handle cases where the model exhibits high confidence yet produces consistent but incorrect reasoning, which is a common issue in RL-trained models?
>
> We thank the reviewer for highlighting the critical issue of overconfidence in RL-trained models. We explicitly address this "consistent but incorrect" failure mode through our active learning sampling strategy and training pipeline:
>
> 1. **Sampling Strategy (Mitigation)**: As mentioned in Line 209, we do not rely solely on uncertainty sampling. In every active learning round, we strategically sample a proportion of high-confidence (high-consistency) data points for expert review. This ensures that cases where the model is "confidently wrong" are captured rather than ignored.
>
> 2. **Correction via Expert Iteration**: Once these high-confidence errors are identified and annotated by experts, they become high-value training signals. In the subsequent Expert Iteration stage, the model is explicitly trained to reject these specific reasoning patterns. By forcing the model to align its high-confidence predictions with expert ground truth, we effectively penalize the "confident hallucination" behavior and recalibrate the model's internal confidence estimates.

---

> ### Author Response · Authors · 2025-12-03
> **Response to Question2 about Ranking Implementation**
>
> > (2) Best-of-N ranking (Section 4.2, L402): Could the authors clarify how ranking is determined? Is OPV used directly to score and rank the outputs through the index of the first incorrect answer?
>
> We appreciate the opportunity to clarify the ranking mechanism used in Section 4.2. The ranking is indeed determined by the OPV's verification verdicts, specifically using the pass rate as the scoring metric.
> Here is the precise procedure:
>
> 1. **Input:** For a given problem, the policy model generates $N$ candidate reasoning trajectories.
> 2. **Verification:** The OPV verifies each candidate trajectory $M$ times. For each verification run, the OPV outputs an index, where \box{STEP-1} indicates a fully correct solution.
> 3. **Scoring:** We calculate a **verification pass rate** for each trajectory, defined as the proportion of the $M$ runs in which the OPV judged the solution as correct (i.e., output \box{STEP-1}).
>   - Score = (Count of \box{STEP-1}) / M
> 4. **Ranking & Selection:** The $N$ trajectories are ranked based on this score. The Best-of-N strategy selects the trajectory with the highest verification pass rate as the final answer.
>
> This voting-based scoring mechanism (aggregating $M$ verdicts) helps robustly estimate the correctness probability, smoothing out the variance of single-pass verification.

---

### Official Review · Reviewer_WDkH · 2025-10-31

**Soundness:** 2
**Presentation:** 3
**Contribution:** 2
**Rating:** 6
**Confidence:** 3

**Summary:**

This work tackles the problem of verifying thoughts (CoTs) generated by models along with the final answers. Given that a large portion of the difficulty in verifying long CoTs comes from the verbosity and noisiness of CoTs, the authors propose to summarize CoTs by keeping only the critical steps that are relevant to the final answer and introduce outcome-based process verifiers (OPV). As this can effectively reduce the size of the inputs to the verification problem, they make use of a human-in-the-loop framework to gather human annotations in stages to improve their OPV models. By demonstrating the performance of the OPV models on ProcessBench and the newly introduced OPV-Bench, they suggest that the proposed approach to summarizing CoTs may make utilizing human annotations for improving the verification performance more viable.

**Strengths:**

1. Reasonable methodological approach
As the complexity of problems increases, the difficulty of verifying thought processes (CoTs) rises, as well. Therefore, linearization and simplification of long CoTs can make the verification problem much smaller and easier, and the proposed summarization approach sounds like a fair approach in this line.

2. Human-in-the-loop pipeline for practical applicability
Taking advantage of the simplified verification process thanks to the summarization, the authors propose an iterative framework to update and improve the verifier model with human interventions. This may encourage practical adoption of the proposed approach by demonstrating how cost-effective human annotations can help improve the verifier.

3. Comprehensive experiments and analyses
This work presents a comprehensive set of experimental results. It covers a fair number of baseline verifiers as well as evaluations on one existing and one in-house benchmarks. Additionally, it provides further analyses and statistics, which can give insights from different angles.

**Weaknesses:**

1. Dependency of OPV on summarization accuracy
Viewing the originally generated CoTs (+ the final answers) as inputs, this work ultimately proposes to factor a process-based verifier into two components: (a) the summarizer and (b) the summarized process-based verifier. However, this work primarily focuses on (b). While the authors stress the importance of the correct summarization and mention re-summarization, it remains at a qualitative level. Importantly, it looks that the summarized CoTs are kept the same for the entire process, making only (b) the target of the updates/optimization. This can pose a concern that this work tackles a subproblem of the original problem (i.e., optimizing (b)) with an assumption of the existence of a reasonable summarizer (i.e., (a)).

2. Potential subjectivity in determining outcome-relevant steps
Depending on the domain, problem, or specific CoT, there may be some subjectivity in whether each step should be considered a "key step" toward the final answer and kept through the summarization. For instance, some steps may not be directly related to the final answer but are still part of the needed exploration to find the correct path to the final answer. Given that this work suggests verifying each of the key steps that are retained in the summarized CoTs, this potential subjectiveness may affect whether each generation should be considered correct or not.

**Questions:**

1. Can you include the ProcessBench performance numbers with OPV but without fine-tuning using human annotations? That could provide a better picture of how the summarization helps with verification.

---

> ### Author Response · Authors · 2025-12-03
> **Response to Weakness1 about Summarization Accuracy**
>
> > 1. Dependency of OPV on summarization accuracy Viewing the originally generated CoTs (+ the final answers) as inputs, this work ultimately proposes to factor a process-based verifier into two components: (a) the summarizer and (b) the summarized process-based verifier. However, this work primarily focuses on (b). While the authors stress the importance of the correct summarization and mention re-summarization, it remains at a qualitative level. Importantly, it looks that the summarized CoTs are kept the same for the entire process, making only (b) the target of the updates/optimization. This can pose a concern that this work tackles a subproblem of the original problem (i.e., optimizing (b)) with an assumption of the existence of a reasonable summarizer (i.e., (a)).
>
> We understand the concern that relying on a static summarizer might reduce the problem to a sub-task dependent on the quality of (a). However, we argue that our approach is robust and practical for the following reasons:
>
> 1. **Summarization is a tractable prerequisite compared to Verification**. Qualitatively, the "summarization" task (removing redundancy) is significantly easier than the "verification" task (detecting logical fallacies). The verbosity in current long reasoning models (like the R1 family) exhibits distinct, recognizable patterns — such as self-correction loops, repeated arithmetic checks, and trial-and-error backtracking. Filtering these out relies largely on semantic pattern matching and information compression, tasks where current SOTA models excel. Furthermore, we employ DeepSeek-V3, a model with strong general-purpose capabilities, combined with refined prompt engineering (as detailed in Appendix C) to ensure high-fidelity summaries.
>
> 2. **The Verifier is trained to be robust to imperfect summaries**. We explicitly designed the optimization of the verifier (b) to adapt to the output distribution of the summarizer (a), rather than assuming (a) is perfect. Crucially, we did not optimize the two components in isolation. Our human annotation protocol (Appendix A.1) serves as a bridge between them. We specifically instructed experts to tolerate "minor logical gaps" and "harmless redundance" in the summaries. The verifier learns to verify the underlying logic even when the summarization is slightly lossy. Consequently, the OPV is not brittle; it does not fail the moment a summary is imperfect. It learns to operate effectively within the "noise" introduced by the summarizer.
>
> We acknowledge that in extreme cases, a "hallucinated summary" could mislead the verifier. In future work, we plan to fine-tune a specialized **Reasoning Summarizer** jointly with the verifier to further minimize information loss.

---

> ### Author Response · Authors · 2025-12-03
> **Response to Weakness2 about Subjectivity in Retaining Steps**
>
> > 2. Potential subjectivity in determining outcome-relevant steps
> Depending on the domain, problem, or specific CoT, there may be some subjectivity in whether each step should be considered a "key step" toward the final answer and kept through the summarization. For instance, some steps may not be directly related to the final answer but are still part of the needed exploration to find the correct path to the final answer. Given that this work suggests verifying each of the key steps that are retained in the summarized CoTs, this potential subjectiveness may affect whether each generation should be considered correct or not.
>
> We appreciate this thoughtful comment regarding the potential subjectivity in summarization. We mitigate this concern through the inherent nature of mathematical reasoning and our robust annotation protocol:
>
> 1. **Logical structure defines relevance**. Unlike open-ended generation tasks, mathematical reasoning relies on strict logical dependencies (i.e., Step B implies Step C). Therefore, the criteria for "key steps" are objectively determined by whether a step serves as a necessary premise for the final conclusion, significantly minimizing subjectivity.
>
> 2. **Robust annotation protocol**. Our annotation guidelines (Appendix A.1) serve as a safety net. Even if the summarizer retains a step that is "exploratory" or "redundant" (subjectively considered non-key), our protocol instructs annotators—and consequently the OPV—not to penalize it. We strictly restrict error flagging to logical fallacies. Thus, the potential inclusion of benign redundant steps does not affect the correctness of the verification.

---

> ### Author Response · Authors · 2025-12-03
> **Response to Question1 about Performance Without Fine-Tuning**
>
> > 1. Can you include the ProcessBench performance numbers with OPV but without fine-tuning using human annotations? That could provide a better picture of how the summarization helps with verification.
>
> We appreciate the suggestion to isolate the impact of our fine-tuning process. We would like to clarify that the R1-Distill-Qwen-32B entry in Table 1 already serves as the performance baseline for the OPV model without our human-in-the-loop fine-tuning (using the same prompt engineering approach).
>
> By comparing this base model against our final OPV-32B, we can clearly observe the performance gains attributed to our training framework:
>
> - On ProcessBench the Rough F1 score improves from 91.1 (Base) to 93.8 (OPV), a gain of +2.7.
> - On OPV-Bench the improvement is even more significant, with the Rough F1 score rising from 75.5 (Base) to 83.1 (OPV), a gain of +7.6.
>
> These results demonstrate that while the base model possesses initial verification capabilities, our iterative fine-tuning on summarized CoTs yields substantial and consistent improvements, particularly on the more challenging OPV-Bench.

---

### Official Review · Reviewer_XyCa · 2025-11-01

**Soundness:** 2
**Presentation:** 2
**Contribution:** 2
**Rating:** 2
**Confidence:** 4

**Summary:**

This paper proposes OPV, which verifies reasoning by first using DeepSeek-V3 to summarize long CoTs into concise solution paths, then performing step-by-step verification on the summaries.

An iterative active learning framework selects uncertain cases for expert annotation. The authors collect 40k annotated solutions and create OPV-BENCH (2.2k samples).

OPV (32B) reportedly outperforms larger models and improves policy model performance on AIME2025 from 55.2% to 73.3% accuracy.

**Strengths:**

1. Clear Problem Motivation: The paper identifies real limitations of OV (ignores process) and PV (expensive for long CoTs), motivating the need for alternative approaches.

2. Substantial Data Collection Effort: Curating 40k expert-annotated solutions with rigorous three-expert consensus protocol represents significant engineering work.

**Weaknesses:**

### 1. Unsubstantiated Efficiency Claims

Total cost = Summarization (671B DeepSeek-V3) + Verification (32B OPV)

Paper provides ZERO computational cost measurements: no latency, FLOPs, or token counts. It claims "efficient" repeatedly but likely more expensive than vanilla PV. This undermines the entire motivation

### 2. Unfair Experimental Comparisons

- Baselines: Qwen3-Max, R1, etc. use zero-shot prompting
- OPV: Uses 40k expert annotations + iterative training + RL

This compares "supervised training with massive expert data" vs. "no training," NOT verification paradigms

Missing critical ablations:
- (1) Same 40k data training full-CoT PV (the real baseline)
- (2) Same 40k data training simple OV
- (3) Summarization contribution quantification (0%? 50%? Unknown)

### 3. Strong Evidence of Data Leakage

Firstly, OPV-BENCH construction and train/test split completely unspecified. The paper says: "Curated from widely-used benchmarks", but  which ones? What overlap?

Secondly, Suspicious performance pattern:
- ProcessBench: OPV 94.4 vs. gpt-oss-120b 93.5 (marginal +0.9)
- OPV-BENCH: OPV 86.2 vs. gpt-oss-120b 77.8 (huge +8.4)

### 4. Active Learning Theatre

Zero experiments comparing active learning vs. random sampling at equal budgets. No justification for this specific uncertainty metric.

**Questions:**

1. If DeepSeek-V3 can perfectly extract key steps, why not use it directly as the verifier?
2. Why does OPV only dominate on its own benchmark? Likely overfitting/leakage

---

> ### Author Response · Authors · 2025-12-03
> **Response to Weakness1 about Unsubstantiated Efficiency Claims**
>
> > Total cost = Summarization (671B DeepSeek-V3) + Verification (32B OPV)
> >
> > Paper provides ZERO computational cost measurements: no latency, FLOPs, or token counts. It claims "efficient" repeatedly but likely more expensive than vanilla PV. This undermines the entire motivation
>
> We appreciate the reviewer’s scrutiny regarding the computational cost. We acknowledge that we did not explicitly breakdown the token metrics in the main text. Below, we provide a detailed analysis demonstrating that our OPV paradigm is indeed more efficient, both computationally and operationally, than a standard Process Verifier (PV) that operates on the full chain-of-thought (CoT).
> 1. **Theoretical Efficiency: Prefilling vs. Decoding.** The reviewer posits that Total Cost = Summarization (671B) + Verification (32B) might exceed that of a Vanilla PV. However, this overlooks the fundamental difference between prefilling (processing input) and decoding (generating output) in LLMs. The computational cost of prefilling (input processing) is significantly lower than that of decoding (output generation) in Decoder-only LLM architectures.
>
> - **OPV Paradigm.** We use DeepSeek-V3 to summarize the raw CoT. While V3 is large (671B), the summarization task is a standard instruction-following task, not a long-reasoning task. It ingests the long raw CoT (prefill, parallelizable and cheap) and outputs a concise rationale (decoding, short and fast). The OPV-32B then takes this short summary and performs rigorous reasoning. Because the input context is significantly compressed, the verifier focuses its "thinking" tokens only on core logical steps.
>
> - **Vanilla PV Paradigm.** To achieve comparable performance, a vanilla LRM-based PV must verify the entire raw CoT. Since raw CoTs contain extensive redundancy (trial-and-error, self-corrections), a Vanilla PV must generate verification reasoning for a massive number of steps. This results in a prohibitive decoding cost, as generation scales linearly with the number of target steps.
>
> 2. **Empirical Evidence: 10x Compression Rate** To quantify this, we analyzed the token usage on OPV-Bench. We randomly sample 500 cases from the OPV-Bench and compare the raw CoTs (inputs for Vanilla PV) against our summarized rationales (inputs for OPV).
>
> **Tokens & Reasoning Steps for Raw CoTs & Summarized Rationales**
>
> | Input | Avg. Tokens | Avg. Steps|
> | :--- | :--- | :--- |
> | Raw CoT | 6785.0 | 72.2 |
> | Summarized Rationale | 1137.8 | 7.0 |
>
> - Long-Reasoning Models (LRMs) solve hard problems by extending their thought process (often 16k-32k tokens). Our summarizer achieves an information compression ratio of roughly 10:1. If a verifier assigns equal computational effort (reasoning tokens) per step, a Vanilla PV would need to generate $\sim$10x more output tokens than OPV. In our preliminary experiments, Vanilla PVs frequently hit context length limits, which is why they were not included as a primary baseline.
>
> - Efficiency is ultimately defined by the trade-off between cost and performance. Our framework allows a small model to outperform a massive one. On OPV-Bench, our 32B model achieves an F1 score of 74.7, significantly outperforming the massive 671B DeepSeek-R1 (F1: 64.7, +10.0 gain).
>
> - Finally, efficiency extends to the human-in-the-loop annotation. Annotating a concise, well-structured rationale (7 steps) is cognitively manageable for experts. In contrast, verifying a raw, meandering CoT (72 steps with backtracks) is prohibitively expensive and error-prone. Our approach makes high-quality process annotation economically viable.

---

> ### Author Response · Authors · 2025-12-03
> **Response to Weakness2 about Unfair Experimental Comparisons**
>
> > - Baselines: Qwen3-Max, R1, etc. use zero-shot prompting
> > - OPV: Uses 40k expert annotations + iterative training + RL
> > This compares "supervised training with massive expert data" vs. "no training," NOT verification paradigms
> Missing critical ablations:
> > - (1) Same 40k data training full-CoT PV (the real baseline)
> > - (2) Same 40k data training simple OV
> > - (3) Summarization contribution quantification (0%? 50%? Unknown)
>
> We understand the reviewer's concern that comparing a supervised model (OPV) against zero-shot baselines might obscure the source of the gains. However, we respectfully argue that this comparison highlights the central value proposition of our work: OPV is not just a model architecture, but an enabling framework that makes large-scale expert annotation feasible.
>
> 1. Feasibility of Ablation (1): Full-Chain PV with Matched Data
>
> The reviewer suggests training a Full-CoT PV on the same 40k annotated dataset as a "real baseline." We emphasize that this ablation is practically infeasible due to the prohibitive cost of annotation and computation for long-reasoning tasks.
>
> - The paradigmatic shift introduced by summarization is intrinsically coupled with the feasibility of supervised training. As noted in our efficiency analysis in Response to Weakness1, raw Long-CoT trajectories average $\sim$72 steps (vs. $\sim$7 steps for OPV).
>
> - Collecting 40k expert annotations on raw CoTs is economically impossible. The cognitive load for experts to pinpoint errors in raw, meandering CoTs is exponentially higher than in concise rationales. We cannot conduct this ablation because the dataset for a "Supervised Full-CoT PV" cannot be constructed at scale. The fact that OPV can be trained on 40k samples is exactly the advantage of our method, as it unlocks a data scale that is inaccessible to standard PVs.
>
> 2. Feasibility of Ablation (2): Training a Simple OV
>
> Regarding OVs, they are intrinsically incapable of step-level verification on benchmarks like ProcessBench, as they only assess final answer correctness against a given ground truth. Our analysis reveals that 30.6% of solutions in ProcessBench and 32.0% in OPV-Bench are "false positives" (correct final answer but flawed reasoning). OVs inherently fail to detect these cases, rendering them unsuitable for this specific evaluation.

---

> ### Author Response · Authors · 2025-12-03
> **Response to Weakness3/Question2 about Data Leakage**
>
> > Firstly, OPV-BENCH construction and train/test split completely unspecified. The paper says: "Curated from widely-used benchmarks", but which ones? What overlap?
> >
> > Secondly, Suspicious performance pattern:
> > - ProcessBench: OPV 94.4 vs. gpt-oss-120b 93.5 (marginal +0.9)
> > - OPV-BENCH: OPV 86.2 vs. gpt-oss-120b 77.8 (huge +8.4)
> >
> > 2. Why does OPV only dominate on its own benchmark? Likely overfitting/leakage
>
> We appreciate the reviewer's scrutiny regarding data hygiene. We assure you that OPV-BENCH was constructed with a rigorous decontamination pipeline to prevent data leakage. (We have also supplemented Appendix A.1 with the following details.)
>
> 1. **Data Sources and Filtering** Our initial data pool was aggregated from three primary sources:
>   - Challenging Benchmarks (~1k): High-difficulty problems from Putnam and USAMO.
>   - Math Forums (~10k): Community-curated advanced problems from AoPS.
>   - Open-Source Datasets (~1000k): Large-scale collections including Numina-Math and AoPS-Instruct.
>
> 2. **Filtering Criteria** First we remove multi-modal queries (i.e., those involving images). Then we explicitly excluded multiple-choice, fill-in-the-blank, and True/False questions, as correct answers for these can often be obtained via "lucky guessing" or shortcuts. We retained only open-ended calculation and proof-based problems that necessitate rigorous, step-by-step reasoning.
>
> 3. **Rigorous Decontamination Pipeline** To prevent data leakage, we implemented a strict decontamination process against major public benchmarks (GSM8K, MATH, OlympiadBench, Omni-MATH, AIME 2024, and AIME 2025). We employed a two-stage filtering mechanism:
>   - **Exact Matching:** Normalized string matching to remove duplicates.
>   - **Semantic Matching:** LLM-based embedding similarity to identify and remove queries that were semantically similar to those benchmarks.
>
> We acknowledge the discrepancy between the marginal gain on ProcessBench (+0.9) and the substantial gain on OPV-BENCH (+8.4). This difference is not due to data leakage, but rather **label noise and saturation within ProcessBench**, specifically regarding the inconsistent treatment of "minor errors" (e.g., slips that are later corrected or do not invalidate the reasoning).
>
> To investigate this, we collected 200 cases where either OPV or gpt-oss-120b disagreed with the ProcessBench ground truth and manually re-annotated a random sample of 50. Our findings revealed significant inconsistency in the benchmark's standards:
>
> * **Inconsistent Standards:** Problems containing similar minor errors were treated differently. For example, *OlympiadBench-532*(the index of the ProcessBench) was penalized as incorrect, while *Math-643* (containing a nearly identical minor error) was ignored and marked correct.
>
> * **Noise Statistics:** Among the 50 sampled problems, 11 (22%) contained minor errors subject to arbitrary penalization, 17 (34%) contained significant reasoning errors that were previously ignored, and 1 was corrupted.
>
> Beyond this noise, ProcessBench exhibits clear saturation, as noted in our paper. This creates an artificial "ceiling," making it difficult for models to surpass ~95% rough accuracy regardless of their true capability. Since the non-controversial problems are already saturated, the remaining "headroom" consists largely of these inconsistent edge cases. This combination of saturation and label noise compresses the performance gap between models. This explains why most models in Table 1 plateau at >90% rough accuracy but struggle to improve further.
>
> In contrast, **OPV-BENCH** features harder problems with explicit, consistent annotation standards for minor errors. This allows it to better distinguish the verification capabilities of stronger models, resulting in the observed +8.4 improvement.

---

> ### Author Response · Authors · 2025-12-03
> **Response to Weakness4 about Active Learning**
>
> > Zero experiments comparing active learning vs. random sampling at equal budgets. No justification for this specific uncertainty metric.
>
> We appreciate the reviewer’s concern regarding the lack of a random sampling baseline and the justification for our Active Learning design. We respectfully argue that a direct comparison with random sampling was not conducted because our objective is practical resource efficiency under strict constraints, rather than proposing a novel AL algorithm. As detailed in Section 2.2, fine-grained process annotation is extremely costly and requires high-level mathematical expertise. Random sampling inevitably wastes this scarce budget on "trivial" cases that the model already solves correctly, yielding low information gain. Our AL strategy is designed to concentrate these limited expert resources solely on "hard" cases where the model exhibits high epistemic uncertainty, thereby maximizing the marginal performance gain per annotation dollar. Thus, we prioritized optimizing the model's performance under a fixed budget over characterizing the theoretical gap between AL and random sampling.
>
> It's important to clarify that Active Learning serves purely as an implementation mechanism for the expert iteration pipeline, enabling the scalable collection of high-quality annotation. Since we do not claim methodological novelty in the field of Active Learning itself, the application and justification of the method are sufficient, and we do not require a separate theoretical comparison against random sampling.
>
> To address your concern regarding the specific parameters and to improve reproducibility, we have expanded Appendix A.1 to include the exact configuration of our sampling strategy.
>
> Our selection strategy balances uncertainty reduction with distribution maintenance:
>
> - **Uncertainty Thresholds**: We set dynamic thresholds for the consistency score to filter uncertain cases. The threshold was set to 0.25 for Stage 1 and increased to 0.5 for Stage 2 as the model became more capable.
> - **Sampling Ratio**: To mitigate the risk of overconfidence or catastrophic forgetting of "easy" modes, we do not sample exclusively from uncertain cases. In each round, 80% of the annotation budget is allocated to the most uncertain cases (below the threshold), while the remaining 20% is randomly sampled from high-confidence cases.

---

> ### Author Response · Authors · 2025-12-03
> **Response to Question1 about Verifier Choice**
>
> > 1. If DeepSeek-V3 can perfectly extract key steps, why not use it directly as the verifier?
>
> We appreciate this insightful question. While DeepSeek-V3 is an exceptionally capable general-purpose model, we chose to separate the pipeline into "Summarization by V3" and "Verification by OPV" based on distinct cognitive demands and empirical performance.
>
> 1. **Distinction in Cognitive Demands**
>
> - **Summarization:** The summarization task in our framework primarily involves identifying and pruning redundant patterns (e.g., self-corrections, trial-and-error loops). This is effectively a semantic pattern-matching and information compression task. DeepSeek-V3, as a powerful generalist LLM, excels at this instruction-following task.
> - **Verification:** Process verification requires rigorous deductive reasoning to pinpoint subtle logical fallacies within a proof. Recent research highlights that specialized Long Reasoning Models (LRMs) (e.g., the R1 family, gpt-oss) significantly outperform standard LLMs on such intensive reasoning tasks. Therefore, relying on V3 directly for verification would be suboptimal. Instead, we choose to fine-tune a specialized verifier based on an LRM foundation (R1-Distill-Qwen-32B).
>
> 2. **Empirical Evidence** LRMs Outperform V3 on Verification. To confirm this, we evaluated DeepSeek-V3 directly as a verifier using the same prompts (see Table 1). The results show that V3 consistently underperforms compared to LRM-based verifers like R1-Distill-Qwen-32B. For example, DeepSeek-V3 achieves a Precise F1 of 72.1, which notably lags behind DeepSeek-R1 at 83.2.
>
> These results confirm that while V3 is an effective summarizer, it lacks the specialized reasoning depth required for state-of-the-art verification. By building OPV on top of an LRM, we achieve superior verification performance.

---

### Meta-Review · Area_Chair_FKjr · 2025-12-18

**Summary:**

Reviewers generally agree that this paper focuses on an important problem: how to verify long CoT reasoning reliably and at lower cost. The proposed outcome-based process verifier (OPV), which summarizes CoT and verifies only the compressed reasoning, is seen as a practical and well-motivated idea. Reviewers found the overall framework intuitive, the motivation convincing, and the empirical results strong, particularly the improvements on ProcessBench and OPV-Bench (developed by the authors), as well as the demonstrated cost reductions.

However, several reviewers raised concerns about experimental clarity and fairness. These include questions about efficiency claims (e.g., what exactly is counted as cost), missing or unclear baselines (especially comparisons to existing PRM/PV methods), limited ablations, and incomplete details about dataset construction and evaluation protocol. While the empirical gains seem promising, it was initially hard to disentangle how much of the improvement comes from summarization, active learning, additional annotations, or verifier design choices.

The rebuttal was detailed and partially addressed these reviewers' concerns with additional explanations, clarifications, and new results. Nevertheless, since the rebuttal was posted quite late (on 3 Dec, which is close to deadline), the reviewers have no chance to provide further feedback.

**Reviewer Concerns:**

### Concerns addressed by the rebuttal: ###

- The authors clarified what is meant by efficiency, breaking down cost into summarization vs verification, explaining prefill vs decode costs, and providing concrete token-level comparisons.

- The authors explained why some zero-shot or non-verification baselines were used and why certain supervised baselines are impractical at OPV scale. Additional results and clarifications helped justify the experimental setup.

- The authors provide a detailed description of data sources, filtering, and decontamination procedures.

- Questions about why DeepSeek-V3 is not used directly as a verifier and how ranking is implemented were answered.

- Additional results showing OPV gains even without human fine-tuning strengthened the practical relevance of the approach.

### Concerns still outstanding: ###

- While partially justified in the rebuttal, the lack of direct head-to-head comparisons with some existing PRM/PV methods under comparable conditions may still lead to confounding conclusions.

- Despite additional discussion, it remains somewhat hard to cleanly separate the contributions of summarization, active learning, and additional annotation effort.

- Reviewers would still prefer broader evaluation beyond ProcessBench-style tasks to better assess generality.

**Reviewer Scores:**

- Reviewer k3ry (Initial score: 4), likely to remain at 4, or possibly increase slightly, as the rebuttal clarifies several experimental and comparison issues. However, concerns about baseline coverage and attribution of gains likely remain.

- Reviewer WDkH (Initial score: 6), likely to remain at 6. The rebuttal addresses most technical questions regarding evaluation protocol, dataset construction, and verifier design, but broader concerns about completeness and scope limit further score increase.

- Reviewer XyCa (Initial score: 2), unlikely to change. Despite detailed rebuttal responses, this reviewer appears to hold fundamental concerns about experimental design, efficiency claims, and overall framing that are not fully resolved.

- Reviewer CTy4 (Initial score: 6), likely to remain at 6, with improved clarity after the rebuttal on baselines, uncertainty handling, and evaluation setup, but without a decisive shift in overall assessment.

---

### Decision · Program_Chairs · 2026-01-26

Reject